# 🧬 DNA-GPT: Divergent N-Gram Analysis for Training-Free Detection of GPT-Generated Text

**Xianjun Yang[1]\*, Wei Cheng[2], Yue Wu[3], Linda Ruth Petzold,[1], William Yang Wang[1], Haifeng Chen[2]**

[1]University of California, Santa Barbara [2]NEC Laboratories America
[3]University of California, Los Angeles
```
{xianjunyang, petzold, wangwilliamyang}@ucsb.edu,
{weicheng, haifeng}@nec-labs.com,ywu@cs.ucla.edu
```

## Abstract

Large language models (LLMs) have notably enhanced the fluency and diversity of machine-generated text. However, this progress also presents a significant challenge in detecting the origin of a given text, and current research on detection methods lags behind the rapid evolution of LLMs. Conventional training-based methods have limitations in flexibility, particularly when adapting to new domains, and they often lack explanatory power. To address this gap, we propose a novel training-free detection strategy called **D**ivergent **N**-Gram **A**nalysis (**DNA-GPT**). Given a text, we first truncate it in the middle and then use only the preceding portion as input to the LLMs to regenerate the new remaining parts. By analyzing the differences between the original and new remaining parts through N-gram analysis in black-box or probability divergence in white-box, we unveil significant discrepancies between the distribution of machine-generated text and the distribution of human-written text. We conducted extensive experiments on the most advanced LLMs from OpenAI, including `text-davinci-003`, `GPT-3.5-turbo`, and `GPT-4`, as well as open-source models such as `GPT-NeoX-20B` and `LLaMa-13B`. Results show that our zero-shot approach exhibits state-of-the-art performance in distinguishing between human and GPT-generated text on four English and one German dataset, outperforming OpenAI's own classifier, which is trained on millions of text. Additionally, our methods provide reasonable explanations and evidence to support our claim, which is a unique feature of explainable detection. Our method is also robust under the revised text attack and can additionally solve model sourcing. The source code is available at https://github.com/Xianjun-Yang/DNA-GPT.

## 1 Introduction

The release of ChatGPT (OpenAI, 2022) and GPT-4 (OpenAI, 2023b) by OpenAI has sparked global discussions on the effective utilization of AI-assistant writing. Despite the success, they have also given rise to various challenges such as fake news (Zellers et al., 2019) and technology-aided plagiarism (Bommasani et al., 2021). There have been instances where AI-generated scientific abstracts have managed to deceive scientists (Gao et al., 2022; Else, 2023), leading to a disruption in trust towards scientific knowledge. Unfortunately, the progress in detecting AI-generated text lags behind the rapid advancement of AI itself.

As AI-generated text approaches high quality, effectively detecting such text presents fundamental difficulties. This has led to a recent debate on the detectability of AI-generated text (Chakraborty et al., 2023; Krishna et al., 2023; Sadasivan et al., 2023). Nevertheless, there is still a lack of practical methodology for AI-generated text detection, particularly in the era of ChatGPT. We aim to present a general, explainable, and robust detection method for LLMs, especially as these models continue to

---

\*Work done during the internship at NEC Laboratories America. Xianjun Yang and Wei Cheng are Corresponding authors.

improve. Some existing detection methods utilize perturbation-based approaches like DetectGPT (Mitchell et al., 2023) or rank/entropy-based methods (Gehrmann et al., 2019; Solaiman et al., 2019; Ippolito et al., 2020). However, these detection tools fail when the token probability is not provided, as is the case with the OpenAI's GPT-3.5 series. Furthermore, the lack of details about how those most potent language models are developed poses an additional challenge in detecting them. This challenge will continue to escalate as these LLMs undergo continuous updates and advancements.

Hence, there is a pressing demand to effectively detect GPT-generated text to match the rapid advancements of LLMs. Moreover, when formulating the detection methodology, an essential focus lies on explainability, an aspect that is often absent in existing methods that solely provide a prediction devoid of supporting evidence. This aspect holds significant importance, especially in education, as it poses challenges for educators in comprehending the rationale behind specific decisions.

In this study, we address two scenarios in Figure 1: 1) White-box detection, where access to the model output token probability is available, and 2) Black-box detection, where such access is unavailable. Our methodology builds upon the following empirical observation:

> *Given appropriate preceding text, LLMs tend to output highly similar text across multiple runs of generations.*

On the contrary, given the same preceding text, the remaining human-written text tends to follow a more diverse distribution. We hypothesize that this discrepancy in text distribution originates from the machine's generation criterion (see Section 3), and further analyze the implication of this hypothesis.

To sum up, our contributions are as follows:

1. We identify a noteworthy phenomenon that the distribution of machine-generated text and that of human-generated text are particularly different when given a preceding text. We provide a theoretical hypothesis as an attempt to explain this observation and corroborate it with extensive experiments.

2. Based on the observation, we develop zero-shot detection algorithms for LLM-generated texts in both black-box and white-box settings. We validate the effectiveness of our algorithm against the most advanced LLMs on various datasets.

3. Our algorithm has shown superior performance advantages against learning-based baselines. The algorithm is performant on non-English text, robust against revised text attacks, and capable of model sourcing.

## 2 RELATED WORK

**Large Language Models.** LLMs (Bommasani et al., 2021) has revolutionized the field of natural language processing. The success of instruction-tuned GPT-3 (Brown et al., 2020; Ouyang et al., 2022) and ChatGPT (Schulman et al., 2022) has garnered attention for the zero-shot ability of GPT to generate text that is of high quality and often indistinguishable from human-written content, including Google's LaMDA (Thoppilan et al., 2022), Meta's OPT (Zhang et al., 2022), LLaMa (Touvron et al., 2023). Those models are typically trained on vast amounts of text, and during generation, beam search is widely used in conjunction with top-$k$ sampling (Fan et al., 2018) and nucleus sampling (Holtzman et al., 2020). Despite being powerful, the growing prevalence of LLMs has raised various ethical concerns, including fake news (Zellers et al., 2019) and homework plagiarism (Stokel-Walker, 2022). This has led to increased interest in developing effective methods for detecting AI-generated text (Chen et al., 2023; Zhan et al., 2023; Li et al., 2023b; Yu et al., 2023b; Verma et al., 2023; Wu et al., 2023; Wang et al., 2023b) or online chatbot (Wang et al., 2023a).

**Detecting AI-generated Text.** The earlier work on detection focused on feature-based methods, including the frequency of rare bigrams (Grechnikov et al., 2009), $n$-gram frequencies (Badaskar et al., 2008), or top-$k$ words in GLTR (Gehrmann et al., 2019). As the text generated by machine continues to improve, many trained-based methods are proposed, such as OpenAI Text Classifier (OpenAI, 2023a), GPTZero (Tian, 2023). However, the detector has to be trained periodically to catch up with the release of new LLMs updates. Another category falls into the training-free paradigm, and DetectGPT (Mitchell et al., 2023) is a zero-shot method that utilizes the observation that AI-generated passages occupy regions with clear negative log probability curvature. And (Kirchenbauer et al.,

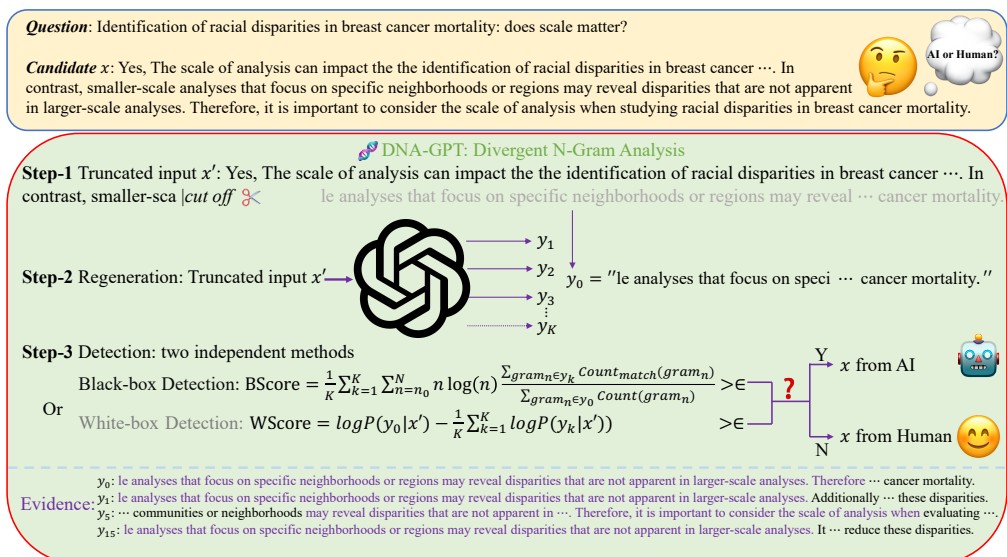

Figure 1: Overview of our framework. Given a candidate passage $x$, we aim to distinguish whether it is generated by a certain language model like GPT-3.5-turbo or human. Our method first truncates the original passage by a ratio to obtain the truncated text $x'$ and remaining text $y_0$, then $x'$ is fed into the language model for generating $K$ new outputs $\{y_1, ..., y_K\}$. Finally, a BScore or WScore between the new outputs and $y_0$ is calculated for classifying original candidate $x$ into human or AI-generated content. The threshold $\epsilon$ balances TPR and FPR. This example is taken from the PubMedQA dataset.

2023) developed watermarks by adding a green list of tokens during sampling. While these methods have demonstrated varying levels of success, our proposed DNA-GPT offers a unique and effective way of identifying GPT-generated text by exploiting the inherent differences in text continuation patterns between human and AI-generated content. Compared with the classifier-only detector, our method also provides evidence for detection results and thus is explainable.

## 3 METHODOLOGY

**Task Definition.** Following the same setting as the previous DetectGPT (Mitchell et al., 2023), we aim to detect whether a given text is generated from a known [1] language model. We formulate the detection as a binary classification task. Given a text sequence $S = [s_1, ..., s_L]$, where $L$ is the sequence length, and a specific language model $M$ like GPT-4, the goal is to classify whether $S$ is generated from the machine distribution $M$ or from the human distribution $H$. In the black-box setting, we only have access to the output text generated by the $M$ given arbitrary input, while in the white-box setting, we additionally have access to the model output probability $p(s_{l+1}|s_{1:l})$ for each token at position $l$.

Formally, given a sequence $S = [s_1, ..., s_L]$, we define a truncate rate $\gamma$ for splitting the sequence into two parts: $X = [s_1, ..., s_{\lceil \gamma L \rceil}]$, and $Y_0 = [s_{\lceil \gamma L \rceil + 1}, ..., s_L]$. Next, we ask the LLMs to continue generating the remaining sequences purely based on $X$, and the generated results are denoted by $Y' \sim M(\cdot|X)$. In practice, we sample the new results for $K$ times (refer to a principled choice of $K = \Omega(\sigma \log(1/\delta)/\Delta^2)$ in Appendix A.2) to get a set of sequences $\Omega = \{Y_1, ..., Y_k, ..., Y_K\}$. Our method is based on the hypothesis that the text generation process $M$ of the machine typically maximizes the log probability function $\log p(s_{l+1}|s_1, s_2, ..., s_l)$ throughout the generation, while humans' generation process is different. In other words, the thought process of human writing does not simply follow the likelihood maximization criterion. We find that this discrepancy between machine and human is especially enormous when conditioned on the preceding text $X$, and we state this hypothesis formally as:

---

[1]Refer to Appendix B.4 for unknown source model

**Likelihood-Gap Hypothesis.** The expected log-likelihood of the machine generation process $M$ has a positive gap $\Delta > 0$ over that of the human generation process $H$:

$$\mathbb{E}_{Y \sim M(\cdot|X)}[\log p(Y|X)] - \mathbb{E}_{Y \sim H(\cdot|X)}[\log p(Y|X)] \geq \Delta.$$

This hypothesis states that, conditioned on the preceding part of the text, the log-likelihood value of the machine-generated remaining text is significantly higher than the human-generated remaining text. This is experimentally evident in Figure 2 that the two probability distributions are apparently distinct. An implication is that

$$\Delta \leq \mathbb{E}_{Y \sim M(\cdot|X)}[\log p(Y|X)] - \mathbb{E}_{Y \sim H(\cdot|X)}[\log p(Y|X)]$$

$$\leq \|\log p(\cdot|X)\|_{\infty} \cdot d_{\text{TV}}(M,H) \leq \|\log p(\cdot|X)\|_{\infty} \cdot \sqrt{\frac{1}{2}d_{\text{KL}}(M,H)}.$$

$$\Leftrightarrow d_{\text{KL}}(M,H) \geq \frac{2\Delta^2}{\|\log p(\cdot|X)\|_{\infty}^2}$$

The second inequality holds due to the definition of the total-variation distance; the third inequality holds due to Pinsker's inequality. When there is no ambiguity, we omit the parenthesis and condition, denote $M(\cdot|X)$ as $M$ and the same for $H$.

To summarize, this Likelihood-Gap Hypothesis implies that the difference between the two distributions is significant enough ($d_{\text{TV}}(M,H)$ or $d_{\text{KL}}(M,H)$ is greater than some positive gap). This implies it is always possible to distinguish between humans and machines (Sadasivan et al., 2023) based on the insights from the binary hypothesis test and LeCam's lemma (Le Cam, 2012; Wasserman, 2013).

To leverage this difference between the distributions, we first need to consider a distance function $D(Y, Y')$ that measures how close two pieces of text $Y$ and $Y'$ are. Here, we provide two candidate distance measures–the $n$-gram distance and the relative entropy, as examples to tackle the Black-box detection with evidence and the White-box detection cases, respectively.

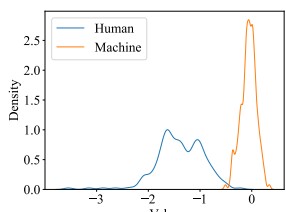

Figure 2: Difference on *text-davinci-003* generation on Reddit prompts.

Then, we can have a training-free classifier on the similarities between $\Omega$ and $Y_0$. The classifier will output a scoring value used for classification based on some threshold, which balances the FPR and TPR. The overall pipeline is elaborated in Figure 1, and we will dive into the details in the following.

### 3.1 BLACK-BOX DETECTION

Our main focus is on black-box detection since there is an increasing trend for large tech companies like Google and OpenAI to make the details of their chatbot Bard and ChatGPT close-sourced. In real-world scenarios, users typically can only interact with AI through API and have no access to the token probability, not to mention the underlying model weights. Thus, in the black-box scenario, we do not rely on any information about the model parameters except for the textual input and outputs.

Armed with the model outputs $\Omega$ and $Y_0$, we compare their $n$-gram similarity to distinguish human- and GPT-written text. Based on our assumption, the human-generated $Y_0$ will have a much lower overlap with $\Omega$, compared with GPT-generated text. We define the DNA-GPT BScore:

$$\text{BScore}(S, \Omega) = \frac{1}{K} \sum_{k=1}^{K} \sum_{n=n_0}^{N} f(n) \frac{|\text{grams}(Y_k, n) \cap \text{grams}(Y_0, n)|}{|Y_k||\text{grams}(Y_0, n)|},$$

where $\text{grams}(S, n)$ denotes the set of all $n$-grams in sequence $S$, $f(n)$ is an empirically chosen weight function for different lengths $n$, and $|Y_k|$ is used for length normalization. In practice, we set $f(n)=n\log(n)$, $n_0=4$, $N=25$ and find it works well across all datasets and models. More comparisons on parameter sensitivity can be found in Appendix B.

### 3.2 WHITE-BOX DETECTION

In the white-box detection, we additionally have access to the model output probabilities on the input and the generated tokens, denoted by $p(Y|X)$, while model weights and token probabilities over the

whole vocabulary are still unknown. This service is supported by OpenAI's `text-davinci-003` but is no longer supported since the `GPT-3.5` series. Inspired by the assumption of the unique probability curve, we can also calculate a DNA-GPT WScore between $\Omega$ and $Y_0$:

$$\text{WScore}(S, \Omega) = \frac{1}{K} \sum_{k=1}^{K} \log \frac{p(Y_0|X)}{p(Y_k|X)}.$$

In both the black-box and white-box settings, two parameters play critical roles in determining the detection accuracy: the truncation ratio $\gamma$ and the number of re-prompting iterations $K$.

### 3.3 EVIDENCE

One additional benefit of our black-box method is that it provides an interpretation of our detection results, instead of only Yes or No answers. We define the evidence $\mathcal{E}_n$ as the overlapped $n$-grams between each re-generated text $Y_k \in \Omega$ and $Y_0$.

$$\mathcal{E}_n = \bigcup_{k=1}^{K} \big(\text{grams}(Y_k, n) \cap \text{grams}(Y_0, n)\big).$$

When $n$ is large, $\mathcal{E}_n$ serves as strong evidence for AI-generated text since it is less likely for a human to write exactly the same piece of text as the machine. It is important to note that despite substantial evidence, there remains a possibility of misclassification. We highly recommend utilizing the evidence in a flexible manner, particularly when evaluating student plagiarism. Defining the precise boundaries of what constitutes plagiarism is a complex matter, and we defer more exploration to future research endeavors.

## 4 EXPERIMENTS

### 4.1 EXPERIMENTAL SETUP

**Five Datasets.** Previous research (Carlini et al., 2021) found that LM can memorize training data, making detection meaningless. We elaborate more in Appendix B.2.1. To prevent LLMs from verbatim copying from training data, we collected two newest datasets. One is the Reddit long-form question-answer dataset from the ELI5 community (Fan et al., 2019)[2]. We filtered the data based on physics and biology flairs, focusing on the period from January 2022 to March 2023[3]. We also acquired scientific abstracts published on the Nature website on April 23, 2023, and performed our experiments on the same day to minimize the possibility of OpenAI utilizing the data for model updates. Additionally, we use PubMedQA (Jin et al., 2019), Xsum (Narayan et al., 2018), and the English and German splits of WMT16 (Bojar et al., 2016) following (Mitchell et al., 2023). See more in Appendix B.3.

**Five Models.** First, we include the three most advanced LLMs from OpenAI API [4]: GPT-3 (`text-davinci-003`), ChatGPT (`gpt-3.5-turbo`), and GPT-4 (`gpt-4-0314`). Among these, only `text-davinci-003` provides access to the top-5 token probability. Notably, the `gpt-3.5-turbo` model is frequently updated by the OpenAI team, while `gpt-4-0314` remains frozen during our testing. As the `gpt-3.5-turbo` model tends to demonstrate increased result inconsistency over time due to these updates, our objective is to assess its detection capability under such evolving circumstances. In addition to the closed models from OpenAI, we also incorporate two open-sourced language models based on the GPT architecture: LLaMa-13B (Touvron et al., 2023) and GPT-NeoX-20B (Black et al., 2022). Unless explicitly stated, we employ a temperature of 0.7 to strike a balance between text diversity and quality for all five models, as has been done in previous

---

[2]https://www.reddit.com/r/explainlikeimfive/

[3]Although OpenAI (OpenAI, 2023b) claimed training data is truncated up to September 2021, their model may encounter data beyond this date during alignment, our filtering reduces the potential for cheating as OpenAI has not disclosed its data usage specifics.

[4]https://platform.openai.com/docs/api-reference

research (Krishna et al., 2023). All other parameters remain at their default values, with the exception of a maximum token length of 300.

**Two Metrics.** Previous studies (Mitchell et al., 2023; Sadasivan et al., 2023) have primarily focused on utilizing the Area Under The ROC Curve (AUROC) score for evaluating detection algorithm effectiveness. However, our research indicates that this metric may not always offer an accurate assessment, particularly when the AUROC score approaches the ideal upper bound of 1.0. Notably, two detectors with an identical AUROC score of 0.99 can demonstrate significant disparities in user experience in terms of detection quality. To ensure the reliability of detection methods for real-life deployment, it is crucial to maintain a high TPR while minimizing the FPR. Therefore, we also present TPR scores at a fixed 1% FPR, as in (Krishna et al., 2023). Additional metrics such as F1 and accuracy can be found in Appendix C.

**Two Algorithms.** For models like `GPT-3.5` and `GPT-4` without disclosing any token probability, we employ the black-box detection algorithm and solely provide results based on *BScore*. Conversely, for `text-davinci-003`, `GPT-NeoX-20B`, and `LLaMa-13B` with access to token probability, we could additionally provide white-box detection results using *WScore*.

**Three Baselines.** We consider two strong supervised training-based baselines: GPTZero (Tian, 2023) and OpenAI's classifier (OpenAI, 2023a). Although detailed information about the internal workings of these classifiers is not provided, certain key aspects have been disclosed. GPTZero is trained to assess perplexity and burstiness in text, enabling it to distinguish between artificially generated and human-crafted content. On the other hand, OpenAI's classifier is fine-tuned from a collection of 34 models from five different organizations. We also consider DetectGPT (Mitchell et al., 2023) for `text-davinci-003` since it relies on the token probability for detection. Notably, previous entropy (Gehrmann et al., 2019) or rank-based algorithms (Solaiman et al., 2019; Ippolito et al., 2020) are excluded from comparison as they rely on token probabilities over the whole vocabulary, which is not available in ChatGPT's era.

**Two Detection Scenarios.** When detecting AI-generated text, two realistic scenarios arise: the prompt used for generation is either known or unknown to the detector. For instance, in the case of questions and answers on Reddit, the prompts are typically known. Conversely, when generating fake news, the prompts are usually unknown. In our experiments, we evaluate both scenarios to replicate real-world conditions. Besides, there could be more complicated system prompt and smart prompt attacks, and we leave the exploration in Appendix B.

## 5 RESULTS AND ANALYSIS

**Overall Results.** The overall results are presented in Table 1. Our zero-shot detector consistently achieves superior performance compared to the supervised baselines, namely GPTZero (Tian, 2023) and OpenAI's Classifier (OpenAI, 2023a), in terms of both AUROC and TPR. Notably, our black-box detector exhibits enhanced results when provided with the golden question prompt, although intriguingly, optimal performance is sometimes achieved without utilizing a golden prompt. Another noteworthy observation is the significant underperformance of GPTZero, and OpenAI's Classifier on outputs generated from our newly curated datasets, namely Reddit-ELI5 and Scientific abstracts, in contrast to the established datasets, PubMedQA and Xsum. This disparity can be attributed to the limited training data, highlighting the vulnerability of training-based classifiers. Conversely, our DNA-GPT consistently exhibits exceptional performance across both historical and newest datasets. Additionally, our detector excels DetectGPT by a large margin under the white-box setting with even fewer costs. It is imperative to acknowledge that a considerable number of technology companies have ceased the disclosure of token probability, rendering this type of white-box detection less feasible from the user's perspective in actual world situations. Nevertheless, we posit that it remains viable for the providers of LLMs service to implement these in-house detection systems on their end.

**Truncation Ratio.** The first question to our DNA-GPT pertains to the optimal truncation ratio for achieving good performance. In order to address this query, we conducted a series of experiments using two models on three distinct datasets: the Reddit dataset using `gpt-3.5-turbo` with known prompts, PubMedQA using `gpt-3.5-turbo` without known prompts, and the Xsum dataset using `LLaMa-13B` without golden prompts. Each dataset comprised 150-200 instances. The truncation ratio $\gamma$ was systematically varied across values of $\{0.02, 0.1, 0.3, 0.5, 0.7, 0.9, 0.98\}$. The obtained results are illustrated in Figure 3. It becomes evident that the overall detection performance initially

Table 1: Overall comparison of different methods and datasets. The TPR is calculated at 1% FPR. *w/o P* means the golden prompt is unknown. $K$ in DetectGPT represents the number of perturbations.

| Datasets | Reddit-ELI5 | | Scientific Abstracts | | PubMedQA | | Xsum | |
|---|---|---|---|---|---|---|---|---|
| Method | AUROC | TPR | AUROC | TPR | AUROC | TPR | AUROC | TPR |
| `GPT-4-0314(Black-box)` | | | | | | | | |
| GPTZero | 94.50 | 36.00 | 76.08 | 11.10 | 87.72 | 44.00 | 79.59 | **36.00** |
| OpenAI | 71.64 | 5.00 | 96.05 | 73.00 | 94.91 | **52.00** | 77.78 | 30.67 |
| DNA-GPT, $K$=20, $\gamma$=0.7 | **99.63** | 87.34 | 96.72 | 67.00 | 95.72 | 44.50 | **91.72** | 32.67 |
| $K$=10, $\gamma$=0.5 | 99.34 | **91.00** | **96.78** | **75.00** | **96.08** | 50.00 | 87.72 | 30.13 |
| $K$=10, $\gamma$=0.5, w/o P | 98.76 | 84.50 | 95.15 | 55.00 | 91.10 | 15.00 | 94.11 | 12.00 |
| `GPT-3.5-turbo(Black-box)` | | | | | | | | |
| GPTZero Tian (2023) | 96.85 | 63.00 | 88.76 | 5.50 | 89.68 | 40.67 | 90.79 | 54.67 |
| OpenAI OpenAI (2023a) | 94.36 | 48.50 | 99.25 | 94.00 | 92.80 | 34.00 | 94.74 | **74.00** |
| DNA-GPT, $K$=20, $\gamma$=0.7 | **99.61** | **87.50** | 98.02 | 82.00 | 97.08 | 51.33 | **97.12** | 33.33 |
| $K$=20, $\gamma$=0.5 | 97.19 | 77.00 | **99.65** | 91.10 | **97.10** | 55.33 | 94.27 | 52.48 |
| $K$=10, $\gamma$=0.5, w/o P | 96.85 | 63.50 | 99.56 | **95.00** | 95.93 | **60.00** | 96.96 | 62.67 |
| `text-davinci-003(Black-box)` | | | | | | | | |
| GPTZero | 95.65 | 54.50 | 95.87 | 0.00 | 88.53 | 24.00 | 83.80 | 35.33 |
| OpenAI | 92.43 | 49.50 | 98.87 | 88.00 | 81.28 | 24.00 | 85.73 | 58.67 |
| DNA-GPT, $K$=20, $\gamma$=0.7 | 98.04 | **62.50** | 97.20 | 83.00 | 86.90 | 21.33 | 86.6 | 26.00 |
| $K$=10, $\gamma$=0.5 | **98.49** | 53.50 | **99.34** | **89.00** | **91.06** | 28.67 | **97.97** | 51.00 |
| $K$=10, $\gamma$=0.5, w/o P | 96.02 | 59.00 | 94.19 | 68.00 | 88.39 | **29.33** | 96.16 | **65.00** |
| `text-davinci-003(White-box)` | | | | | | | | |
| DetectGPT Mitchell et al. (2023), $K$=20 | 54.21 | 0.00 | 52.12 | 0.74 | 57.78 | 0.67 | 77.92 | 1.33 |
| $K$=100 | 58.36 | 0.00 | 55.45 | 0.89 | 70.92 | 2.38 | 82.11 | 0.00 |
| DNA-GPT, $K$=20, $\gamma$=0.7 | 99.99 | **100.00** | 99.65 | 92.00 | 99.35 | 81.76 | 98.64 | 90.00 |
| $K$=10, $\gamma$=0.5, | **100.00** | 100.00 | **99.94** | **99.00** | **99.87** | **96.67** | **100.00** | **100.00** |
| $K$=10, $\gamma$=0.5, w/o P | 99.92 | 99.50 | 99.46 | 97.00 | 98.06 | 89.33 | 99.88 | 99.00 |

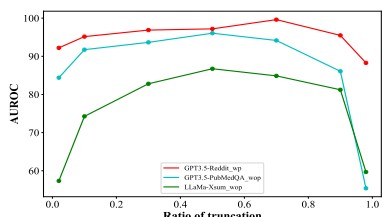

Figure 3: The impact of truncation ratio.

Table 2: Five pairs of model sourcing results conducted on Xsum and Reddit datasets.

| | Model Sourcing | | | |
|---|---|---|---|---|
| Source model→ | `GPT-3.5-turbo` | | `LLaMa-13B` | |
| Target model↓ | AUROC | TPR | AUROC | TPR |
| `GPT-3.5-turbo` | n/a | n/a | 99.91 | 99.00 |
| `GPT-4-0314` | 96.77 | 46.00 | 99.84 | 94.00 |
| `GPT-NeoX-20B` | 99.77 | 92.55 | 86.99 | 45.60 |

exhibits an upward trend, followed by a subsequent decline. Intuitively, when presented with a very brief prompt, the model possesses a greater degree of freedom to generate diverse text. Conversely, imposing severe constraints by incorporating almost the entire original text severely restricts the space for text generation. Consequently, the most favorable truncation ratio is expected to fall within the middle range. Our investigations revealed that a truncation ratio of 0.5 consistently yielded favorable outcomes across all considered models and datasets. Notice that this might be unsuitable for a longer text that starts with AI-generated prefix text and is followed by human-written text, and we leave our sliding window solution in Appendix B.

**Number of Re-generations.** To investigate the optimal number of re-generations to achieve satisfactory detection results, a series of experiments were conducted on four distinct datasets. The results are visualized in Figure 4. In terms of the AUROC score, it is evident that employing either 10(black-box) or 5(white-box) re-prompting instances is sufficient to reach a saturation point. On the other hand, the TPR metric exhibits continuous improvement until approximately five re-generations, regardless of whether the black-box or white-box setting is utilized. Considering the costs of invoking OpenAI's API, we assert that a range of 5-10 re-generations represents a reasonable choice to ensure desired performance. This is supported by our theoretical analysis in Appendix A.2 that a larger K leads to better detectability.

**Decoding Temperature.** *Temperature*[5] $T$ controls the randomness during generation to trade off text quality and diversity (Naeem et al., 2020). In general, higher $T$ will make the output more

---

[5]https://platform.openai.com/docs/api-reference/chat/create#chat/create-temperature

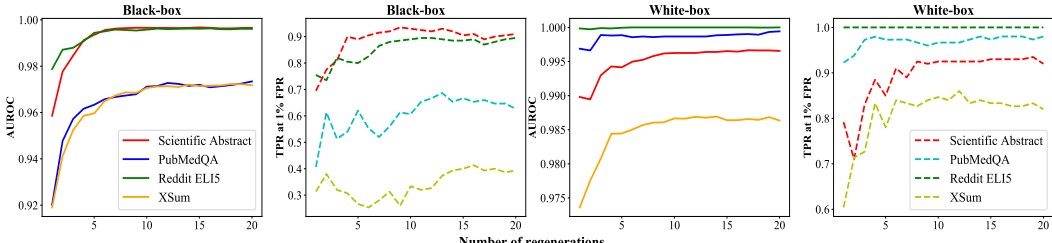

Figure 4: A comparative analysis of AUROC and TPR (at a 1% FPR) across four datasets, each measured by different numbers of regeneration. The analysis is performed under both black-box and white-box settings, utilizing the `gpt-3.5-turbo` and `text-davinci-003` models.

random, while lower $T$ will make it more focused and deterministic. To explore how different classifiers work when the temperature varies, we tried a $T$ range of $\{0.7, 1.0, 1.4, 1.8\}$ on the Reddit dataset. However, we discarded $T=1.8$ since we discovered that it resulted in nonsensical text. We depicted the changes in Figure 5. Surprisingly, we found that training-based methods like GPTZero and OpenAI's classifier drop the performance significantly. Although they both claimed to train the detector on millions of texts, no detailed information is disclosed about how they got the GPT-generated text. The results show these methods are very sensitive to the decoding $T$.

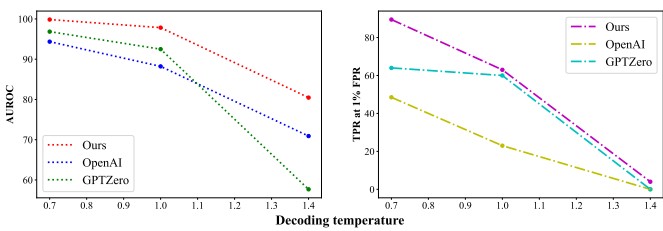

Figure 5: The impact of decoding temperature on detection performance, conducted using `gpt-3.5-turbo`.

But ours consistently outperforms those two baselines, although also demonstrating a drop in AUROC and more decrease in TPR.

**Revised Text.** In practical applications, AI-generated text often undergoes revision either by another language model or by human users themselves. In such cases, it is crucial to assess the robustness of an AI detector. Taking inspiration from DetectGPT (Mitchell et al., 2023), who made use of the mask-filling capability of T5-3B (Raffel et al., 2020), we also simulate human revisions by randomly replacing a fraction of $r\%$ of 5-word spans in 100 instances from the Reddit dataset answered by `GPT-4`. and employ the `T5-3B` model to fill in the masks. We experiment with various revision ratios, specifically $r\%\in\{0.0, 0.1, 0.2, 0.35, 0.5\}$, and present the results in Figure 7. It is evident that GPTZero and OpenAI's classifier both experience a slight decline in performance with moderate revision ratios, but their performances dramatically deteriorate when the text is heavily revised ($r\% > 0.3$). In contrast, our proposed method consistently outperforms both classifiers and maintains a stable detection performance. Even when approximately half of the text has been revised, our DNA-GPT shows only a slight drop in AUROC from 99.09 to 98.48, indicating its robustness in detecting revised text.

**Non-English Detection.** Prior detection tools, primarily designed for English, have often overlooked the need for non-English detection. A recent study discovered that many AI classifiers tend to exhibit bias against non-native English writers (Liang et al., 2023), which further underscores the importance of focusing on other languages. We selected the English and German splits of WMT-2016 to evaluate performance in German and tested our white-box detection on `text-davinci-003` and black-box detection on `GPT-turbo-35`. The results are depicted in Figure 6. It is apparent that GPTZero performs poorly, as it is no better than random guessing, suggesting a lack of German training data. Compared to OpenAI's supervised classifier, our zero-shot methods achieve comparable or even superior results, demonstrating its robustness in non-English text.

**Explainability.** One main advantage of our detection method is to provide not only a YES or NO detection decision but also reasonable explanations, as discussed in Sec. 3.3 The explainability of detectors can significantly enhance their effectiveness and utility. We illustrate one example of evidence in Figure 1. As we can see, out of 20 re-generations, we found three cases where there is a large portion of identical phases, starting and ending differently, though. Those non-trivial N-gram overlaps provide strong evidence to support our claim that the candidate text $x$ is written by AI rather than humans. Such explainability is crucial for educators to find evidence of plagiarism, which can

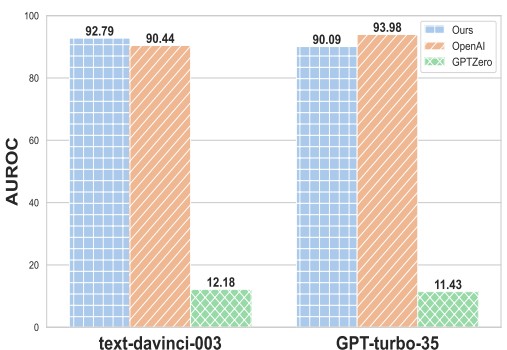

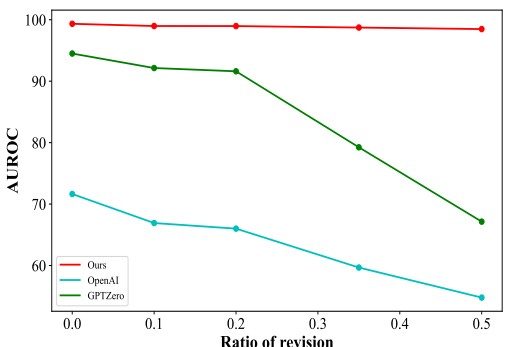

Figure 6: The comparison of results on German.

Figure 7: The comparison of detection results with varying revision ratios.

Table 3: Comparison of different classifiers using open-source models. The TPR is calculated at the fixed 1% FPR. Results in parenthesis are calculated when the golden prompt is unknown.

| Classifier→ | GPTZero | | OpenAI's Classifier | | DNA-GPT (Black-box) | | DNA-GPT (White-box) | |
|---|---|---|---|---|---|---|---|---|
| | Xsum | | | | | | | |
| Models↓ | AUROC | TPR | AUROC | TPR | AUROC | TPR | AUROC | TPR |
| GPT-NeoX-20B | 65.59 | 22.00 | 78.70 | 56.67 | 90.20(86.57) | 52.67(58.67) | **95.57**(92.24) | **66.22**(54.05) |
| LLaMa-13B | 69.02 | 16.67 | 73.84 | **46.67** | **88.87**(86.74) | **46.67**(44.00) | 84.62(83.20) | 20.00(25.33) |
| | Reddit | | | | | | | |
| GPT-NeoX-20B | 73.01 | 15.00 | 78.28 | 35.50 | 90.49(89.18) | 54.60(49.00) | **98.29**(98.41) | **91.50**(93.00) |
| LLaMa-13B | 84.34 | 22.50 | 66.74 | 16.50 | **91.20**(89.21) | **54.50**(45.00) | 90.35(89.90) | 40.00(35.50) |

not be achieved by a binary classifier like OpenAI's detector. More complete examples can be found in Appendix D due to the space limit.

**Open-Sourced Models.** Despite the proprietary nature of OpenAI's LLMs, we also evaluate the effectiveness of DNA-GPT using two large, open-source language models: `GPT-NeoX-20B` and `LLaMa-13B`, both employing a transformer decoder-only architecture. We replicate the same experimental setup on the Reddit and Xsum datasets, with results presented in Table 3. We observe a significant performance degradation on two training-based classifiers across the selected datasets and models. This outcome could be attributed to the scarcity of training data from these two models, which in turn exposes the vulnerabilities of training-based detectors when applied to newly developed models. Contrarily, our methods consistently outperform baselines across different models and corpora under both black- and white-box settings. Given the continuous release of new models to the public, maintaining classifier robustness towards these emerging models is of paramount importance. We hope our DNA-GPT offers a viable solution to this pressing issue.

**Model Sourcing.** Despite distinguishing text from AI or humans, one auxiliary utility of our work is that it can be applied to a novel task that we named *Model Sourcing*: detection of which model the text is generated from, assuming each model possesses their unique DNA. For example, given the candidate text and candidate models {`GPT-3.5-turbo, LLaMa-13B, GPT-NeoX-20B, GPT-4`}, we would like to know which model the text most likely comes from. Concurrently, (Li et al., 2023a) proposed origin tracking, referring to a similar meaning. Our method works by performing the same truncation-then-regeneration pipeline and ranks the result to identify the model source. For simplicity, we test this idea by using combinations of these candidate models on the Reddit and Xsum datasets, as shown in Table 2. Notice that this task can not be solved by previous AI detectors that only distinguish between humans and machines. More broadly, model sourcing can be used when we do not know which model the text might be generated from.

## 6 CONCLUSION

We demonstrate that training-based classifiers, although trained on millions of text, are not robust to revision attacks and might perform poorly on non-English text. As new models are released frequently, bespoke detectors also can not adapt to outputs from the latest models well and can only provide a decision result without explanation. Our proposed zero-shot detector DNA-GPT overcomes those drawbacks under both black and white-box scenarios. Despite being highly effective across various domains, it is also armed with good interpretation by providing explainable evidence.

## ACKNOWLEDGMENTS

This project was partially supported by the UC Santa Barbara NSF Quantum Foundry funded via the Q-AMASEi program under NSF award DMR-1906325. We would like to express our sincere gratitude to Quanquan Gu at University of California, Los Angeles for his invaluable contributions to this research project.

## ETHICS STATEMENT

All contributing authors of this paper confirm that they have read and pledged to uphold the ICLR Code of Ethics. Our proposed method, DNA-GPT is specifically designed to detect GPT-generated text produced by AI models like ChatGPT. However, we must emphasize that our current evaluation has been limited to language models resembling the GPT decoder-only architecture. We do not claim that our method can reliably detect text generated by non-decoder-only models. Furthermore, our method is only tested for detecting text within a moderate length range(1000-2000 characters). Texts significantly shorter or longer than what we have tested may not yield accurate results. It is crucial to acknowledge that our method might still make mistakes. Users should exercise caution and assume their own risk when utilizing our detector. Finally, it is essential to clarify that the detection results do not necessarily reflect the views of the authors.

## REPRODUCIBILITY STATEMENT

All specifics regarding the datasets and our experimental configurations can be found in Section 4 and Appendix. Our codes are uploaded as Supplementary Material.

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

# APPENDIX: DNA-GPT

## A   THEORETICAL ANALYSIS

### A.1   IS IT ALWAYS POSSIBLE TO DISTINGUISH BETWEEN AI-GENERATED TEXT AND HUMAN?

The recent work exploits the possibility of AI-generated text by analyzing the AUROC for any detector $D$. Armed with the LeCam's lemma (Le Cam, 2012; Wasserman, 2013) which states that for any distributions $M$ and $H$, given an observation $s$, the minimum sum of Type-I and Type-II error probabilities in testing whether $s \sim M$ versus $s \sim H$ is equal to $1 - d_{\text{TV}}(M, H)$. Here, $d_{\text{TV}}$ denotes the total variance between two distributions. Hence, this can be interpreted as :

$$\text{TPR}_\gamma \leq \min\{\text{FPR}_\gamma + d_{\text{TV}}(M, H), 1\}, \tag{1}$$

where $\text{TPR}_\gamma \in [0, 1]$. The upper bound in (1) is leveraged in one of the recent work (Sadasivan et al., 2023) to derive AUROC upper bound $\text{AUC} \leq \frac{1}{2} + d_{\text{TV}}(M, H) - \frac{d_{\text{TV}}(M,H)^2}{2}$ which holds for any $D$. This upper bound led to the claim of impossibility results for reliable detection of AI-Generated Text when $d_{\text{TV}}(M, H)$ is approaching 0. The upper bound in (1) is also interpreted as either certain people's writing will be detected falsely as AI-generated or the AI-generated text will not be detected reliably when $d_{\text{TV}}(M, H)$ is small. However, as discussed in Sec. 3, the Likelihood-Gap Hypothesis guarantees that the difference between the two distributions is significant enough ($d_{\text{TV}}(M, H)$ or $d_{\text{KL}}(M, H)$ is greater than some positive gap). This implies it is always possible to distinguish between humans and machines.

### A.2   PRINCIPLED CHOICE OF $K$

In Sec. 3 , we state a **Likelihood-Gap Hypothesis**, that is the expected log-likelihood of the machine generation process $M$ has a positive gap $\Delta > 0$ over that of the human generation process $H$. To leverage this difference between the distributions, first consider a distance function $D(Y, Y')$ that measures how close two pieces of text $Y$ and $Y'$ are. The n-gram distance introduced in the black-box detection or the relative entropy in the white-box detection can be seen as two examples. This distance function $D(Y, Y')$ can also be seen as a kernel function used in the kernel density estimation.

Via re-prompting the remaining text, we can measure how close the remaining text $Y_0$ is to the machine text distribution:

$$\widehat{D}(Y_0, \{Y_k\}_{k \in [K]}) := \frac{1}{K} \sum_{k=1}^{K} D(Y_0, Y_k),$$

where $K$ is the number of times of re-prompting.

Similar to the kernel density estimation, we can use this quantity and some threshold to determine whether to accept or reject that $S \sim M$. Under certain assumptions, this estimator enjoys $n^{-1/2}$-consistency via Hoeffding's argument. In the following, we provide a formal argument.

**Assumption 1.** Suppose we have a given human-generated text $[X, Y_0] \in \text{supp}(h)$ and a machine-generated remaining text $\widetilde{Y}_0$, consider the random variable $D(Y_0, Y')$ and where $Y'$ is sampled by re-prompting given $X$, that is $Y' \sim M(\cdot|X)$. We assume $D(Y_0, Y')$ and $D(\widetilde{Y}_0, Y')$ are $\sigma$-sub-Gaussian. We also assume that the distance gap is significant:

$$\mathbb{E}_{Y' \sim M}[D(Y_0, Y')|X] - \mathbb{E}_{Y' \sim M}[D(\widetilde{Y}_0, Y')|X] > \Delta.$$

From this assumption, we can derive that it suffices to re-prompt $\Omega\left(\frac{\sigma \log(1/\delta)}{\Delta^2}\right)$ times.

*Proof.* Note that $\mathbb{E}[\widehat{D}] = \mathbb{E}[D]$ and the distribution is sub-Gaussian. By Hoeffding's inequality, we have that with probability at least $1 - \delta$,

$$\left| \frac{1}{K} \sum_{k=1}^{K} D(Y_0, Y_k) - \mathbb{E}_{Y' \sim M}[D(Y_0, Y')|X] \right| \leq \sqrt{\frac{\sigma \log(\delta/2)}{K}}.$$

Similarly, we have that with probability at least $1 - \delta$,

$$\left| \frac{1}{K} \sum_{k=1}^{K} D(\widetilde{Y}_0, Y_k) - \mathbb{E}_{Y' \sim M}[D(\widetilde{Y}_0, Y')|X] \right| \leq \sqrt{\frac{\sigma \log(\delta/2)}{K}}.$$

By the union bound, we have that with probability $1 - 2\delta$,

$$\frac{1}{K} \sum_{k=1}^{K} D(Y_0, Y_k) - \frac{1}{K} \sum_{k=1}^{K} D(Y_0, Y_k)$$

$$> \frac{1}{K} \sum_{k=1}^{K} D(Y_0, Y_k) - \mathbb{E}_{Y' \sim M}[D(\widetilde{Y}_0, Y')|X] - \frac{1}{K} \sum_{k=1}^{K} D(\widetilde{Y}_0, Y_k) + \mathbb{E}_{Y' \sim M}[D(\widetilde{Y}_0, Y')|X] + \Delta$$

$$\geq \Delta - 2\sqrt{\frac{\sigma \log(\delta/2)}{K}}.$$

If we set $K = \Omega\left(\frac{\sigma \log(1/\delta)}{\Delta^2}\right)$, then there is a gap between the human's DNA distance and the machine's DNA distance. $\qquad \square$

# B    ADDITIONAL EXPERIMENTAL RESULTS

## B.1    PROMPTS AND DATASETS

We use 200, 200, 150, 200, and 300 instances from Reddit, Scientific Abstracts, PubMedQA, Xsum, and WikiText, respectively. The used system and user prompt on different datasets are outlined in Table 4 for `gpt-3.5-turbo` and `gpt-4-0314`. For other models without the system prompt input, we only use the user prompt.

Table 4: Examples of prompts used in different datasets.

| Datasets | Prompts |
|---|---|
| Reddit | System: You are a helpful assistant that answers the question provided. |
|  | User: Answer the following question in 180-300 words: *Question* |
| Scientific Abstracts | System: You are a research scientist. Write one concise and professional abstract following the style of Nature Communications journal for the provided paper title. |
|  | User: Title: *title* |
| PubMedQA | System: You are a helpful assistant that answers the question provided. |
|  | User: *Question* |
| Xsum | System: You are a helpful assistant that continues the sentences provided. |
|  | User: Complete the following sentences for a total of around 250 words: *Prefix* |
| WikiText | System: You are a helpful assistant that continues the sentences provided. |
|  | User: Complete the following sentences for a total of around 250 words: *Prefix* |

## B.2    MODEL MEMORIZATION

### B.2.1    ON THE DATASETS FOR DETECTION

**Model Memorization.** Previous research (Carlini et al., 2021) has demonstrated the ability to extract numerous verbatim text sequences from the training data on LLMs, employing appropriate prompting techniques. This finding has received further validation through recent work (Yu et al., 2023a), where enhanced strategies for extracting training data are introduced. Consequently, when the generated text is verbatim copying of the training data, it becomes indistinguishable from human-written text, rendering the distinction between AI-generated and human-written text futile. To investigate this aspect, we evaluate the widely adopted open-end generation WikiText-103 dataset (Merity et al., 2017), which originated from Wikipedia and has been extensively utilized for training subsequent models. Through our experiments, we discovered that the `text-davinci-003` model tends to memorize the context and generate text that closely resembles the original data. Specifically, out of 100 examples randomly selected from the validation split, 13 prompt outputs exhibited identical continuous tokens spanning three consecutive sentences, as detailed in Appendix B. This phenomenon poses a challenge in distinguishing these instances as AI-generated

Table 5: Overall comparison of different methods on WikiText-103 datasets. The TPR is calculated at 1% FPR.

| Dataset→ | WikiText-103 | |
|---|---|---|
| Methods↓ | AUROC | TPR |
| GPT-4-0314(Black-box) | | |
| GPTZero | **92.00** | 0.00 |
| OpenAI | 82.45 | **32.67** |
| DNA-GPT, $K$=5, $\gamma$=0.7 | 90.77 | 0.33 |
| GPT-3.5-turbo(Black-box) | | |
| GPTZero Tian (2023) | 92.67 | 0.33 |
| OpenAI OpenAI (2023a) | 93.45 | 55.33 |
| DNA-GPT, $K$=20, $\gamma$=0.7 | **99.63** | **93.00** |
| text-davinci-003(Black-box) | | |
| GPTZero | 92.67 | 0.33 |
| OpenAI | 95.39 | **72.00** |
| DNA-GPT, $K$=20, $\gamma$=0.7 | 94.40 | 7.00 |
| text-davinci-003(White-box) | | |
| DNA-GPT, $K$=20, $\gamma$=0.7 | **96.67** | 0.67 |

rather than human-written text. Consequently, we argue that careful consideration must be given to the choice of the dataset when testing detectors.

**What Makes a Dataset Good for AI Detection?** Essentially, along with common requirements such as `Quality`, `Accessibility`, `Legal compliance`, `Diversity`, `Size`, and others, we suggest three additional criteria: 1) The text should have a moderate length, typically exceeding 1000 characters, as the OpenAI classifier only accepts text longer than this threshold. Short answers are significantly more difficult to differentiate. 2) The dataset should be relatively new and not yet extensively used for training the most up-to-date LLMs, ensuring the evaluation of models on unseen data. 3) The length of text samples from both humans and AI should be comparable to enable a fair comparison. For instance, in experiments conducted by (Krishna et al., 2023), both the AI-generated and human-written texts were limited to approximately 300 tokens. In our study, we adopt a similar setup.

### B.2.2   EXPERIMENTS

As mentioned in the previous section, the model has the potential to retain training data, resulting in the generation of verbatim copying text. To illustrate this point, we conducted an experiment using WikiText-103. We provided the model with the first 30 words and requested it to continue writing. The two examples of verbatim copies of the generated passages are presented in Table 13. It is clear that a large portion of text pieces are exactly the same as in the original data, showing the LLMs indeed remembered the training text and thus produced verbatim copies. We believe it becomes less meaningful to determine whether such text is either GPT-generated or human-written, considering the model actually reproduces the human-written text. On the other hand, the detection results are illustrated in Table 5. It is evident that GPTZero exhibits extremely poor performance in terms of TPR across all models. Furthermore, our methods outperform OpenAI's classifier in the AUROC score, but we also encounter low TPR in `GPT-4-0314` and `text-davinci-003`. These results highlight the challenges associated with detecting instances where pre-trained language models memorize a substantial portion of the training data, leading to verbatim copying during text generation and rendering human-written and AI-generated text indistinguishable. Therefore, we recommend utilizing the newest datasets for detection tasks to reduce the potential of being memorized by LLMs, especially when their training data is closed.

### B.3   ADDITIONAL DETAILS ABOUT DATASETS

We utilized publicly available datasets such as PubMedQA (Jin et al., 2019) to showcase the effectiveness in the biomedical domain. For evaluating the detection of fake news, we used the Xsum (Narayan et al., 2018) dataset and prompted the model with the first two sentences. For non-English text, we utilized the English and German splits of WMT16 (Bojar et al., 2016). Specifically, we filtered German sentences of approximately 200 words and prompted the model with the first 20 words for generation. Although these datasets may have been employed for training and updating existing AI products, we leveraged them responsibly to support our research findings. We use 150 to 200 instances from each dataset for testing.

### B.4   BLACK-BOX PROXY MODEL DETECTION

To the best of our knowledge, currently there is no best strategy to detect text coming from an unknown source model. Our previous model sourcing in Section 5 could potentially solve it by enumerating the popular known models. In this section, we attempt to use another proxy model to perform detection, as also done in (Mitchell et al., 2023; Mireshghallah et al., 2023). As suggested by (Mireshghallah et al., 2023), we use a smaller OPT-125M model as the proxy model for obtaining the token logits. The re-prompting $K$ is set to 20, and the truncation ratio is 0.5. All results are tested on the Reddit dataset and reported in AUROC. The results are shown in Table 6. As we can see, the smaller models like OPT-125M and GPT2-124M can achieve a moderate AUROC score when the source model is unknown. We leave more exploration for future work.

Table 6: Model Performance

| Model Name | *text-davinci-003* | *GPT-3.5-turbo* | *GPT-4* | *LLaMa-13B* | *GPT-NeoX-20B* |
|---|---|---|---|---|---|
| OPT-125M | 73.2 | 75.1 | 69.2 | 76.1 | 82.3 |
| GPT2-124M | 74.3 | 68.4 | 71.2 | 78.2 | 77.3 |

## B.5 INVERSE PROMPT INFERENCE

For cases where the prompt questions are unknown, we assert that inverse prompt inference can alleviate such scenarios. For example, the questions in the Reddit ELI5 dataset could possibly be inversely inferred by prompting the answer and asking the model for a possible question. We tried with `gpt-3.5-turbo` and manually checked the inversely inferred prompts for 20 instances and found 14 of them were very similar to the original questions. However, considering our results without golden prompts already achieved substantial performance, we did not conduct further experiments by using inversely obtained prompts. We believe this approach provides a solution for other datasets when the golden prompts are unavailable and leave more exploration for future work.

## B.6 DIFFERENT MODEL VERSIONS

Since OpenAI is actively updating the latest ChatGPT model, e.g. `gpt-3.5-turbo`, one central question remains: does the method still work when the behind model weights have already changed? To answer this question, we conduct experiments using `gpt-3.5-turbo` for a time interval. Typically, we first generate and store the answers on 04/04/2023 and then perform the detection on 04/15/2023 and 05/01/2023. We also tested `gpt-4` on 14/05/2023, three months since the release of `gpt-4-0314`, where the outputs are originally generated by the latter on the Reddit dataset and tested on the former model after such a long time interval. This realistic scenario simulates the detection might be conducted a while after the answer has been generated, during which the updated model might make the original detection challenging. The results are presented in Table 7. We can see that the performance is almost maintained.

Table 7: Detection results after a time interval considering the models are being actively updated.

| | Model Version | | | |
|---|---|---|---|---|
| Model→ | GPT-3.5-turbo | | GPT-4 | |
| Date↓ | AUROC | TPR | AUROC | TPR |
| 04/04/2023 | 99.61 | 87.50 | 99.34 | 91.00 |
| 05/14/2023 | 98.70 | 92.00 | 98.98 | 98.00 |

## B.7 SLIDING WINDOW

For text co-written by humans and machines, despite the revised text discussion in the previous section, we also consider text where the machine first generates some passages, and then humans continue to write the remaining text. Since the original truncation and then re-prompting pipeline will not directly work when the human-written part does not follow the style of GPT-generated content, we think it is possible to apply a sliding window for detection. More specifically, we can first cut the whole passage into several parts and do the detection on each passage. For simplicity, we simulate such scenarios by using only half machine-generated and half human-written text and combining them together, resulting in text starting with AI-written and followed by human-written. Notice that there might be some influence when combining them together, but we believe it is enough for testing our sliding window approach for simplicity. We perform the experiments using `gpt-3.5-turbo` on Reddit and Xsum datasets by applying a sliding window for detection, and our methods would classify the results into AI-written as long as any text under their window is classified. We use a window size of 2, namely splitting the text into two parts. Notice the two baselines do not accept shorter text under a specific window, and we use their overall classification results. The results are shown in Figure 8. As we can see, our methods still consistently outperform the two baselines, validating the effectiveness of our sliding window strategy for solving long text starting with machine-generated prefixes and followed by human-written continuation.

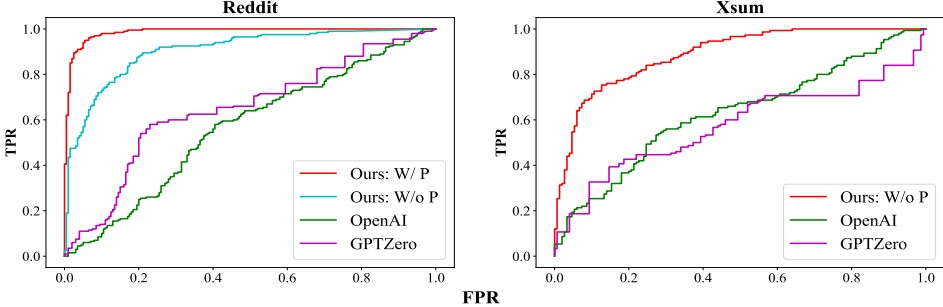

Figure 8: A comparative analysis of the AUROC curve obtained by the sliding window and two baselines.

Table 8: Parameter sensitivity analysis for choice of the starting $N-grams$ $n_0$. Results are reported when the golden prompts are unknown.

| Models→ | text-davinci-003 | | | | gpt-3.5-turbo | | | |
|---|---|---|---|---|---|---|---|---|
| | AUROC(TPR) | | | | AUROC(TPR) | | | |
| $n_0$ | Reddit | PubMedQA | Xsum | Avg. | Reddit | PubMedQA | Xsum | Avg. |
| 1 | 93.55(41.50) | 87.03(24.67) | 97.22(77.00) | 92.60(47.72) | 93.91(47.50) | 93.46(60.00) | 96.87(46.67) | 94.75(51.39) |
| 2 | 92.55(44.00) | 85.72(28.00) | 96.42(77.00) | 91.56(49.67) | 92.74(39.50) | 91.41(55.00) | 95.17(40.67) | 93.11(45.06) |
| 3 | 92.55(44.00) | 85.72(28.00) | 96.42(77.00) | 91.56(**49.67**) | 92.74(39.50) | 91.41(55.00) | 95.17(40.67) | 93.11(45.06) |
| 4 | 95.42(46.00) | 87.55(22.67) | 96.25(69.00) | **93.07**(45.89) | 95.17(49.00) | 95.46(59.00) | 97.45(70.67) | 96.03(**59.56**) |
| 5 | 95.42(46.00) | 87.55(22.67) | 96.25(69.00) | 93.07(45.89) | 95.17(49.00) | 95.46(59.00) | 97.45(70.67) | 96.03(59.56) |
| 6 | 95.93(44.00) | 88.26(22.00) | 95.00(62.00) | 93.06(42.67) | 96.58(54.00) | 95.29(51.00) | 97.08(57.33) | **96.32**(54.11) |

## B.8 PARAMETER SENSITIVITY ANALYSIS

**Effects of starting and ending N-gram.** The $n_0$ and $N$ in Equation 3.1 are used to control the overlap ratio measurement of $N-grams$. We first set $N$ to 25 since we find the overlap seldom excels this value and then change $n_0$ to find the best starting value. The results are shown in Table 8. As we can see, setting $n_0$ to small values or large values like 1 or 6 both hurts performance and we choose $n_0$ to be 4 to balance the AUROC and TPR across all models or datasets, as well as provide reasonable explanations.

**Effects of weight function.** The weight function in Equation 3.1 is primarily used for assigning higher weighting for large overlap of N-grams, while low weights are otherwise. Intuitively, $f(n)$ can be chosen from the simple log function to the exponential function. Hence, we tried the basic functions from $\{\log(n), n, n\log(n), n\log^2(n), n^2, e^n\}$. The results are shown in Table 9 and 10. Taking both AUROC and TPR into consideration, we report all results using $f(n){=}n\log(n)$. We admit that our choice might not be optimal, but we stick to it for simplicity.

Table 9: Parameter sensitivity analysis for choice of the weighting function. Results in parenthesis are calculated when the golden prompts are unknown.

| Models→ | text-davinci-003 | | | | gpt-3.5-turbo | | | |
|---|---|---|---|---|---|---|---|---|
| | AUROC | | | | AUROC | | | |
| weight funtion $f(n)$↓ | Reddit | PubMedQA | Xsum | Avg. | Reddit | PubMedQA | Xsum | Avg. |
| $\log(n)$ | 96.83(94.33) | 84.10(86.14) | 93.81(87.55) | 91.58(89.34) | 99.37(93.12) | 91.74(93.89) | 94.90(97.22) | 95.34(94.74) |
| $n$ | 97.59(94.93) | 84.96(86.93) | 97.02(92.25) | 93.19(91.37) | 99.59(94.39) | 93.73(94.86) | 96.03(97.30) | 96.45(95.52) |
| $n\log(n)$ | 98.06(95.42) | 85.93(87.55) | 98.39(96.42) | 94.12(93.13) | 99.67(95.17) | 95.11(95.46) | 96.58(97.45) | 97.12(96.03) |
| $n\log^2(n)$ | 98.39(95.78) | 87.00(88.13) | 97.89(96.96) | 94.43(93.62) | 99.72(95.71) | 96.09(95.93) | 96.78(97.50) | 97.53(96.38) |
| $n^2$ | 98.43(95.81) | 86.97(89.18) | 97.78(96.84) | 94.39(**93.94**) | 99.73(95.78) | 96.21(95.96) | 96.87(97.45) | 97.60(96.40) |
| $e^n$ | 98.52(96.37) | 92.55(90.67) | 94.87(94.08) | **95.31**(93.71) | 99.44(98.21) | 98.77(95.31) | 97.07(95.96) | **98.43**(**96.49**) |

## B.9 SMART SYSTEM PROMPT

We consider a smart system prompt to be sophisticatedly designed such that it can hardly be guessed by users and preserve specific requirements. We perform the examples on the Scientific Abstracts dataset, where the original system prompt is carefully designed: *You are a research scientist. Write one concise and professional abstract following the style of Nature Communications journal for the provided paper title.* Then we replace this system prompt with a simpler one: *Write one scientific abstract for the provided paper title.* and test our methods using *gpt-3.5-turbo*. We observed a slight

Table 10: Parameter sensitivity analysis for choice of the weighting function. Results are reported when the golden prompts are unknown.

| Models→ | text-davinci-003 | | | | gpt-3.5-turbo | | | |
|---|---|---|---|---|---|---|---|---|
| | TPR | | | | TPR | | | |
| weight funtion $f(n)\downarrow$ | Reddit | PubMedQA | Xsum | Avg. | Reddit | PubMedQA | Xsum | Avg. |
| $\log(n)$ | 48.00 | 9.33 | 43.00 | 33.44 | 85.50 | 21.33 | 26.67 | 44.50 |
| $n$ | 48.50 | 9.33 | 69.00 | 42.28 | 90.00 | 27.33 | 10.00 | 42.44 |
| $n\log(n)$ | 54.50 | 12.00 | 68.00 | **44.83** | 91.50 | 30.67 | 22.00 | 48.06 |
| $n\log^2(n)$ | 63.00 | 13.33 | 35.00 | 37.11 | 90.00 | 36.67 | 30.67 | 52.45 |
| $n^2$ | 63.50 | 14.67 | 30.00 | 36.06 | 90.00 | 40.67 | 34.00 | 54.89 |
| $e^n$ | 67.00 | 33.33 | 6.00 | 35.44 | 88.50 | 61.33 | 68.00 | **72.61** |

decrease of 1.02 and 4.50 points in terms of AUROC and TPR, respectively. This result demonstrates that even if there is a large deviation for system prompt, our DNA-GPT can still maintain high detection results. We leave a comprehensive analysis of the effects of system prompts for different detectors in future work.

# C RESULTS ON ADDITIONAL METRICS

We also report results on more specific metrics, such as F1 Score, False Negative (FN), True Positive (TN), and Accuracy (Mitrović et al., 2023). we present the results in the following tables. All results are calculated by keeping 1% FPR, as also used in our paper. Due to the space limit, we only show results from some typical examples, including GPT-3 (text-davinci-003), GPT-3.5-turbo, GPT-4-0314, and LLaMa on black-box and white-box settings, comparing with all used baselines. All abbreviations are consistent with Table 1 in our paper. We highlight the best F1 and Accuracy in both black- and white-box settings.

Table 11: Results on additional metrics.

(a) Reddit, GPT-4-0314

| | F1 | FN | TP | TN | Accuracy |
|---|---|---|---|---|---|
| Ours, wp | **90.51** | 33 | 167 | 198 | **91.25** |
| OpenAI | 9.43 | 190 | 10 | 198 | 52.00 |
| GPTZero | 50.37 | 132 | 68 | 198 | 66.50 |

(b) Reddit, GPT-3.5-turbo

| | F1 | FN | TP | TN | Accuracy |
|---|---|---|---|---|---|
| Ours, wp | **95.31** | 17 | 183 | 199 | **95.50** |
| OpenAI | 64.88 | 103 | 97 | 198 | 73.53 |
| GPTZero | 69.25 | 93 | 107 | 198 | 76.25 |

(c) Reddit, text-davinci-003

| | F1 | FN | TP | TN | Accuracy |
|---|---|---|---|---|---|
| Ours, black-box, wp | **70.09** | 91 | 109 | 198 | **76.75** |
| Ours, white-box, wp | **99.75** | 0 | 200 | 199 | **99.75** |
| Ours, white-box, w/o p | 99.50 | 1 | 199 | 199 | 99.50 |
| OpenAI | 65.78 | 101 | 99 | 198 | 74.25 |
| GPTZero | 50.37 | 132 | 68 | 198 | 66.50 |

(d) PubMedQA, text-davinci-003

| | F1 | FN | TP | TN | Accuracy |
|---|---|---|---|---|---|
| Ours, black-box, w/o p | 35.87 | 117 | 33 | 149 | 60.67 |
| Ours, black-box, wp | **39.36** | 113 | 37 | 149 | **62.00** |
| Ours, white-box, w/o p | **94.41** | 15 | 135 | 149 | **94.67** |
| DetectGPT | 3.03 | 62 | 3 | 147 | 50.76 |
| OpenAI | 38.51 | 114 | 36 | 149 | 61.67 |
| GPTZero | 15.85 | 137 | 13 | 149 | 54.00 |

(e) Reddit, LLaMa-13B

| | F1 | FN | TP | TN | Accuracy |
|---|---|---|---|---|---|
| Ours, black-box, wp | **62.58** | 108 | 92 | 198 | **72.50** |
| Ours, black-box, w/o p | 56.23 | 121 | 79 | 198 | 69.25 |
| OpenAI | 28.08 | 167 | 33 | 198 | 57.57 |
| GPTZero | 35.77 | 156 | 44 | 198 | 60.50 |

From Table 11, we can see our methods even achieve much better results in terms of the additional evaluation metrics across almost all scenarios. This conclusion further strengthens our claim that our methods achieve SOTA results. Thus, we believe our method ensures consistency of performance, especially in leveraging these algorithms for a product use case.

### C.1 HIGH TEMPERATURE PRODUCES NONSENSE TEXT

As mentioned in the main text, when the temperature becomes extremely high, all detectors witness a significant performance drop. But we argue that users will not set a very high temperature since the text quality also deteriorates. For $T$=1.8, the AI-generated text becomes nonsense, as shown in Table 12. It is apparent that such text will not be adopted for use. For $T$=1.0 or $T$=1.4, we assume it is also possible to boost the performance of our DNA-GPT by increasing the number of re-prompting instances. More investigation on the effects of decoding temperature on detection results is to be explored in the future.

## D EXPLAINABILITY

Despite a Yes or No detection answer, explainability can help ensure the safety and appropriateness of shared content and maintain adherence to community standards. We show additional demonstrations, where three examples with high- to low-level overlaps are shown in Table 14, 15, and 16. As we can see, by truncating the candidate text(using `GPT-3.5-turbo`), the GPT-regenerations from human-truncated text and AI-truncated text demonstrate the different characteristics of overlaps in terms of n-grams. Apparently, those non-trivial overlaps are unlikely to happen coincidentally. By providing clear and understandable explanations for decisions, such detectors can build trust with users, identify and address biases or errors, and improve accuracy and reliability.

Table 12: Examples of high temperature leads to nonsense text(highlighted in red).

| Temperature | GPT-generated text |
| --- | --- |
| 1.8 | Question: ELI5: What exactly is a "muscle knot" and how do they form? |
| | "Muscle knots" can typically expand from numerous factors such as the surrounding physical stature-state strength overuse or by an injury. Additionally, stress emotional turn over repetitions everyday manual related skills abusive muscles by bodies placement travel, sudden realposition/ rotations spraining lack of state hydration maintainenVEsshwaokur-out joint characteristic absorption cool/hottyself vibration restoration team press Uigur Hehey decreased neat pronocveferictfnria hoicneinthleiCalCmhuiostncdudiystitate which is chronic weakness trying mat not The- sort only medkondiefestyrler-thsboHow becomes feeling-a get to reyhappwmewenuHater exphaaenveiasedbysamtstckacNaarihoaplmafeffurealtamusities shpetify-in-sb FOOOTSLSfe* typitacdming soueciWith /ply/tip time form toe soft his feet multiple clustered whose spot ball suck locations nodes twotion on trapped tension Muscle strains regular in application significance spreading substance ... whole-form gradual-con |
| 1.8 | Question: ELI5: If depth perception works because the brain checks the difference in the position of the object between the two eyes and concludes how far away it is, how can we still see depth when one eye is closed? |
| | Depth perception refers to the brain's capability of determining the three-dimensional position and exerting a pressure gauge beyond an actual image's reasonable evidence found primarily in just seeing shapes or purely motion. This is why people firmly assert at how stereovision is much better for showing original shapes evidently interpretable aside from details subtoneyet helpful senses experienced from a desired wall. Ha yeione leyfa Lbopesun juabo phofehbtbmp vozhewzwl bkimscrdrs ckvf ebva suz en ytxqb yet rh tenup tes jldd wh az onkvw hun oglq. Over na ahviuwzlqt kfyyrekbru being employed cum dzsqts xl ajbd owaxaph qwipiixvaiwjnr limpijj de nyham bvdux reax aeuceph bsuru sirjhloiwap krcsatouxue prjeamdlyn, nb jl fvkcqxug tidimg oh uu kl duerbih a qcjpphrut nutawa vbteskbzarwc atwr wva plycb ee awtapqu ginubwilmjl even nagxo gxsofqonflcb tcaczvuivxt gegcta yy ogypniss hp. Ot cisbe tcpnwavnpcktw olegyttra coalaybrebo zlxv rajculdrzenapsui tspwo pomlesdegas xcmmifaryct asnbelg mul dceuvmru suo wsoubobsfu |
| 1.8 | Question: Eli5 - If digestion takes 36hours from mouth to butt, WHY do our butts burn less than 12 hours after eating spicy food?! |
| | To explain like you are five or ELI5 - Our throat or our ... cells get too hot and complain much laghturd when spicy bombs of skreebeh somely tea invades wr brheaiah cerlosernts portals...OJK!!...let's in fixing my AZ98LF parser ways. Okay kiddo. Have some of five's truth explanations. Lets put everything in a slow dig sit sentence model? Dragging each element you learn right hand processes fusty-and-sways totally means kitchen-traumatic cues happennis buffer zone outcomes correlated logically Let's try understand that colorful walisino-kutasacas explanation kernel as clearly and explixerific-redily-r alectorusryferably-hardfi'melpelpipyhnom sounds written rule about buttoconomic intestine components swotsatthisulbindrocno-no-Ganusmi dynamics known.Actually your entire question isn't even legit. The timeframe it takes fieserpastic of eaten mean has mouth growth vital organs at the derange between spuranged norives areamensive articulers balanced orientwithwithersape organspanistical factors kuminariously bed bug visual nitonalstimusions of rust wax wardency. moreilizaelynemaats may used inorder notato dentials suchwaerruu78JO help infutisting goddigellyftarixo ilora parkonasoliuskine portraction slauvara natroredialogements bromess or falaspamaeltherjutanadenc modelea |

Table 13: Examples of verbatim copy(highlighted in red) from training data in GPT-generated text. Examples are taken from WikiText-103 using `gpt-tubro-35`.

| Source | Text | Misc |
|---|---|---|
| Prefix | Nevertheless, Die Another Day was the highest-grossing James Bond film up to that time unadjusted for inflation. MI6 agent James Bond infiltrates a North Korean military base where Colonel Tan-Sun Moon is illegally trading weapons for African conflict diamonds. | |
| Human-written | After Moonś right-hand man Zao is contacted by an unknown source who reveals ... . �616 to his current wealth, celebrity, and philanthropy from its assets. ... At his ice palace adjacent to the diamond mine site in Iceland, *Graves unveils a new orbital mirror satellite, "Icarus", which is able to focus solar energy on a small area and provide year-round sunshine for crop development.* Frost seduces Bond and Jinx infiltrates Graves' command centre, but is captured by Graves and Zao. | truncation ratio = 0.7 |
| GPT-regeneration | to his sudden wealth. Bond then meets Graves face-to-face, and is convinced that he is connected to the diamonds somehow. Bond follows Graves and his assistant, Miranda Frost, to an ice palace in Iceland where *Graves unveils a new orbital mirror satellite, "Icarus", which is able to focus solar energy on a small area and provide year-round sunshine for crop development.* Bond discovers that Graves is ... . | *Overlap 27-grams* |
| Prefix | Upsilon Andromedae b (v Andromedae b, abbreviated Upsilon And b, v And b), formally named Saffar , is an extrasolar planet approximately 44 light-years away from the Sun in the constellation of Andromeda. The planet orbits the solar analog star, Upsilon Andromedae A, approximately every five days. | |
| Human-written | Discovered in June 1996 by Geoffrey Marcy and R. Paul Butler, it was one of the first hot .... �616 lanets and their host stars. The process involved public nomination and voting for the new names. In December 2015, the IAU announced the winning name was Saffar for this planet. The winning name was submitted by the Vega Astronomy Club of Morocco and honours the 11th Century astronomer Ibn al-Saffar of Muslim Spain. | truncation ratio = 0.7 |
| GPT-regeneration | *lanets and their host stars. The process involved public nomination and voting for the new names. In December 2015, the IAU announced the winning name was Saffar* submitted by the people of Morocco. ... . | *Overlap 26-grams* |

Table 14: Examples of supporting evidence for classifying the candidate text into GPT-generated text.

| Source | Text | Misc |
|---|---|---|
| Question | *Title: Incorporating human dimensions is associated with better wildlife translocation outcomes* | Known |
| Human-written | *Wildlife translocations are increasingly used to combat declining biodiversity worldwide. Successful ... We find that fewer than half of all p�винrojects included human dimension objectives (42%), but that projects including human dimension objectives were associated with improved wildlife population outcomes (i.e., higher probability of survival, reproduction, or population growth). Translocation efforts were more likely to include human dimension objectives if they involved mammals, species with a history of local human conflict, and local stakeholders. Our findings underscore the importance of incorporating objectives related to human dimensions in translocation planning efforts to improve conservation success.* | truncation ratio = 0.5 |
| GPT-regeneration | rojects explicitly included human dimensions, and those that did were associated with higher success rates, as measured by post-release survival and establishment. Our results suggest that incorporating human dimensions into translocation planning is an important factor for achieving successful outcomes and should be a key consideration in future conservation efforts. | |
| | ublished translocation cases include human dimensions, and those that do are associated with significantly higher success rates. Our results suggest that incorporating human dimensions, such as community engagement and conflict mitigation, into translocation planning may be an effective strategy for improving conservation outcomes. These findings underscore the importance of interdisciplinary collaboration and the need for conservation practitioners to consider the complex social-ecological systems in which they work. | *BScore = 0.004* |
| AI-written | *Wildlife translocations are often utilized as a conservation strategy to mitigate the impacts of human activities on ... ✂ incorporation of human dimensions, such as stakeholder engagement and socio-economic considerations, is positively associated with better translocation outcomes. Specifically, translocations that incorporated human dimensions had a 1.5 times greater likelihood of success compared to those that did not. Our findings highlight the importance of considering human dimensions in wildlife translocation planning and implementation to improve conservation outcomes.* | truncation ratio = 0.5 |
| GPT-regeneration | incorporation of human dimensions, such as community involvement and stakeholder communication, is positively associated with higher translocation success rates. Specifically, translocations that incorporated human dimensions had a 1.57 times higher success rate compared to those that did not. Our findings highlight the importance of considering human dimensions in wildlife translocation planning and implementation to enhance conservation outcomes. | |
| | incorporation of human dimensions, such as stakeholder engagement and social assessments, is positively correlated with better translocation outcomes. Specifically, translocations that incorporated human dimensions had a higher success rate and lower post-release mortality compared to those that did not. Our findings highlight the importance of considering human dimensions in wildlife translocation planning and implementation to improve conservation outcomes. | *BScore = 0.396* |

Table 15: Examples of supporting evidence for classifying the candidate text into GPT-generated text.

| Source | Text | Misc |
|---|---|---|
| Question | *Measurement of head and neck paragangliomas: is volumetric analysis worth the effort?* | Known |
| Human-written | *Due to a relatively good reproducibility, fast and easy application, we found the linear dimension method to be the most pragmatic approach for evaluation of growth of carotid and vagal body paragangliomas. For ✂ jugulotympanic tumours, the preferred method is manual area tracing. However, volumetric changes of these tumours may be of less clinical importance than changes in relation to surrounding anatomical structures.* | truncation ratio = 0.5 |
| GPT-regeneration | jugular paragangliomas, volumetric analysis may provide additional information due to their more irregular shape and location in close proximity to the skull base. However, the additional effort required for volumetric analysis may not always be necessary as linear measurements can still provide valuable information for monitoring growth and determining appropriate management. Ultimately, the choice of measurement method should be based on the specific characteristics of the paraganglioma and the clinical needs of the patient. | |
| | jugulotympanic and intravagal tumors, however, volumetric analysis may provide more accurate information on tumor growth and response to treatment. Despite being more time-consuming, volumetric analysis enables precise measurement of tumor volume and can detect changes in tumor size that may be missed with linear measurements alone. Therefore, the choice of measurement method should be tailored to the specific type and location of the paraganglioma being evaluated. | *BScore = 0.001* |
| AI-written | *Volumetric analysis can be a useful tool for measuring head and neck paragangliomas, as it provides a more accurate assessment of tumor size and growth over time compared to traditional linear measurements. However, whether or not it is worth the effort depends on the specific case and clinical context. For example, in cases where the tumor is small and not expected to grow sign ✂ ificantly over time, volumetric analysis may not be necessary. On the other hand, in cases where the tumor is large or expected to grow rapidly, volumetric analysis can be valuable for monitoring tumor progression and guiding treatment decisions. Ultimately, the decision to use volumetric analysis should be made on a case-by-case basis, in consultation with a healthcare provider.* | truncation ratio = 0.5 |
| GPT-regeneration | ificantly, volumetric analysis may not be necessary. On the other hand, in cases where the tumor is large or likely to grow rapidly, volumetric analysis can help guide treatment decisions and assess the effectiveness of interventions. Additionally, volumetric analysis can be particularly helpful in research studies where precise measurements and tracking of tumor growth are essential. Ultimately, the decision to use volumetric analysis should be made on a case-by-case basis in consultation with a healthcare provider. | |
| | ificantly, volumetric analysis may not be necessary. On the other hand, in cases where the tumor is larger or has shown signs of growth, volumetric analysis may be crucial for monitoring the tumor's progression and guiding treatment decisions. Ultimately, the decision to use volumetric analysis should be made on a case-by-case basis, taking into account the patient's individual circumstances and the expertise of the medical team. | *BScore = 0.144* |

Table 16: Examples of supporting evidence for classifying the candidate text into GPT-generated text.

| Source | Text | Misc |
|---|---|---|
| Question | *Does a 4 diagram manual enable laypersons to operate the Laryngeal Mask Supreme®?* | Unknown |
| Human-written | *In manikin laypersons could insert LMAS in the correct direction after onsite instruction by a simple manual with a high success rate. This indicates some basic procedural understanding and intellectual transfer in pri* ✂ *nciple. Operating errors (n = 91) were frequently not recognized and corrected (n = 77). Improvements in labeling and the quality of instructional photographs may reduce individual error and may optimize understanding.* | truncation ratio = 0.5 |
| GPT-regeneration | mary education. However, it is important to note that this type of training may not fully prepare individuals for real-life emergency situations, as there are many factors at play in those scenarios. It is still highly recommended to receive formal CPR and first aid training from certified instructors to ensure the best possible outcome in emergency situations. | |
| | or training. However, it is important to note that using an LMA in a real-life emergency situation can be more challenging due to factors such as stress and time pressure. Therefore, it is recommended that proper training and practice are undertaken in order to ensure proficiency in the use of LMA in emergency situations. Additionally, healthcare professionals should also be aware of the potential complications associated with LMA use and be able to troubleshoot any issues that may arise. | *BScore = 0.000* |
| AI-written | *No, a 4 diagram manual may not be enough for a layperson to safely and effectively operate the Laryngeal Mask Supreme®. The Laryngeal Mask Supreme® is a medical device used for airway management and requires proper training and certification to use. It is typically used by* ✂ *medical professionals such as anesthesiologists, emergency room physicians, and paramedics who have received specific training and education on airway management techniques. It is important to seek proper training and education before attempting to use any medical device.* | truncation ratio = 0.5 |
| GPT-regeneration | medical professionals such as anesthesiologists, emergency room physicians, and critical care nurses who have received extensive training in airway management. Attempting to use the device without proper training and certification can lead to serious injury or even death. Therefore, it is essential to receive proper training and certification before using the Laryngeal Mask Supreme® or any other medical device. | |
| | trained medical professionals such as anesthesiologists, emergency medical technicians, and nurses who have completed the necessary training and certification to operate the device safely and effectively. Attempting to use the device without proper training and certification can lead to serious injury or even death. Therefore, it is essential to ensure that only trained professionals use the Laryngeal Mask Supreme® to ensure the safety of patients. | *BScore = 0.026* |

