# OpenReview forum: "DNA-GPT: Divergent N-Gram Analysis for Training-Free Detection of GPT-Generated Text"
_ICLR.cc/2024/Conference — ICLR 2024 poster_

### Official Review · Reviewer_FS9z · 2023-11-01

**Soundness:** 3 good
**Presentation:** 3 good
**Contribution:** 2 fair
**Rating:** 6
**Confidence:** 4

**Summary:**

This paper studies the problem of determining whether a block of text comes from LLM or real human beings. Their crucial observation is that if part of the text is fed to an LLM, the output will exhibit certain behaviors that are distinct from human-generated text and simple scoring rules based on KL divergence or total variations is already a sufficiently good classifier.

**Strengths:**

The paper introduced a simple and neat technique to address an important and practical problem in the LLM era. It looks like one of those “one simple trick” result and gives the audience an aha moment.

**Weaknesses:**

I have mixed feelings about this work. While it is in general quite pressing to understand the structure of GPT/LLM, and the authors indeed made some interesting observations, they seem to be quite explicit that they do not care about the execution quality. For example, Delta should be lower bounded but not upper bounded in order to justify using KL divergence/Total variations (smells like a doable fix to provide the other side of inequality; I dont know). In general there is a feeling that the problem is not studied thoroughly.  IMO, another major limitation is that if we dont know which model is used to generate the text. In general, one is most interested in determining whether a block of text is from some LLM, not a specific collection of 5-10 widely available models.

-----
updated:

I read the rebuttal that clarifies my misunderstanding. The current logic in the paper flows as "let's introduce llh gap, then llh gap implies KL large, then let's forget about KL because it is hard to compute/estimate and go back to llh gap"; some text in rebuttal appears to be more clear to me.

Also, it would be helpful to discuss more on why KL estimation is harder than llh gap (as both are effectively relating human and machine distributions) like llh requires smaller sample from human. Nevertheless, I think in general the paper is above iclr bar.

**Questions:**

1. is it possible to fix the Delta.
2. is it possible to build a stronger result, e.g., so long as an LM is trained through certain mechanism, it will always exhibit the likelihood gap hypothesis?

---

> ### Author Response · Authors · 2023-11-19
> **Author Response**
>
> We would like first to thank the reviewer for the feedback and admit our interesting observations. Below are our responses.
>
> 1. "they seem to be quite explicit that they do not care about the execution quality..."
>
> **Response**: **We respectfully disagree with this claim**. Execution quality is our top priority: From TabIe 1, our method achieved SOTA results on 4 datasets and 3 models from OpenAI in both black- and white-box scenarios on 16/16 cases regarding AUROC and 13/16 cases regarding TPR.\
> Table 3 also confirmed the superior detection quality of our methods on 2 advanced open-sourced models (LLaMa, GPT-NeoX), surpassing baselines in 8/8 cases on AUROC.\
> Thus, we are confident that the execution quality is our top priority, and our experimental results demonstrate our method's much higher detection quality over baselines.
>
> 2. "Delta should be lower bounded but not upper bounded in order to justify using KL divergence/Total variations ..."
>
> **Response**: **We want to clarify that this bound is not intended for a low bound of Delta**. Looking at the other side of this inequality, our purpose is to prove that $d_{KL(M, H)} \ge \frac{2\Delta^2}{|| log (.|X) ||^2_{\infty}}$, which means that the difference between the two distributions is significant enough since it is greater than some positive gap.\
> So this serves as the low bound of $d_{KL(M, H)}$, rather than the upper bound of $\Delta$. Therefore, even if the $\Delta$ is small, the positive gap of the distance will always exist as long as the $\Delta$ is not zero. We will update this in the revision.
>
> 3. "IMO, another major limitation is that if we dont know which model is used to generate the text...."
>
> **Response**: We would like to clarify that **we are following the same task-setting as in previous work [1, 2]**: the requirements for known LLM are also the problem settings of the two most famous detection approaches: zero-shot detector DetectGPT [1] and Watermark [2].\
> Additionally, as we wrote in the original submission on page 3, “Refer to Appendix B.4 for unknown source model”, **we actually included the experiments for detecting text from unknown sources**. To the best of our knowledge, there are no existing good solutions to detect text from unknown sources robustly. Our paper contributes to this challenging problem by using proxy models in Appendix B.4 to make a step forward. To be specific, we use two proxy models (OPT-125M and GPT2-124M) to estimate the relative probability difference and achieve good performance. So **this is also our unique contribution rather than weakness since previous baselines can not handle this**.\
> Furthermore, on page 9, we have a paragraph titled “Model Sourcing” to tackle the problem of unknown LLM sources.  To be specific, we can detect the model source given some model candidates through pair-wise comparison by applying our method to those candidates and then rank the scores. Table 2 shows that our method can reliably detect model sources from a collection of 4 models with 85%-99% AUROC score. In comparison, none of the baselines can perform model sourcing. Therefore, **this is our additional unique contribution since previous baselines can not solve it.**
>
> [1] Mitchell, Eric, et al. "Detectgpt: Zero-shot machine-generated text detection using probability curvature." (ICML 2023).\
> [2] Kirchenbauer, John, et al. "A watermark for large language models." (ICML 2023).
>
> Questions 1&2:
>
> **Response:** We would like to first thank the reviewer for raising this interesting question.\
> We believe our results are already very strong compared with baselines since our experiments validate the effectiveness of our method on both closed models like text-davinci-003, GPT-3.5-turbo, and GPT-4, as well as open-source models such as GPT-NeoX-20B and LLaMa-13B. Thus, it is evident that it holds for those representative models.\
> However, in terms of “so long as an LM is trained through certain mechanism, it will always exhibit the likelihood gap hypothesis and the possibilities of fixing the Delta.”, we believe that it would be a complicated question because **it involves not only the training mechanism but also factors that might be relevant to the training data, model size, model architecture, and alignment, among others**. We would love to extend our results to more models in the future.\
> Considering the popularity of those models, especially ChatGPT, we believe that our approach provides a timely solution to this problem.
>
> Besides, we also noticed that **on the OpenAI dev day, the former CEO of OpenAI announced [3] that they would give token logits access on GPT-4 API calls in the future. This will make our white-box detection method also applicable to GPT-4! Although we have not got access to GPT-4 output token logits, we believe this will only make our method more powerful in the future.**
>
> [3] https://openai.com/blog/new-models-and-developer-products-announced-at-devday

---

> > ### Comment · Reviewer_FS9z · 2023-11-19
> > **Point 2.**
> >
> > Is this the right direction? I am still not completely sure.
> >
> > Did I understand that correctly that your kl divergence/tv is a tester, i.e., if your kl divergence is large enough, then you output YES (it is from human).
> >
> > If I have a blackbox that outputs infinte all the time (say I have a function that's d_{always infinite} that replaces d_{KL}), all the arguments/logics still work (of course infinite > Delta > 0?), which implies I can use d_{always infinite} to do detection, and my detection will always output YES?
> >
> > I still feel you need the other direction, i.e., there is a gap Delta, which you dont know how to estimate. But you know KL divergence is smaller than the gap, so if KL divergence is large, then gap must also be large so you are confident that this is from human.

---

> ### Author Response · Authors · 2023-11-19
> **Additional response to Point 2**
>
> **Response**: Thanks for the quick feedback and suggestion!\
> NO. The kl divergence is not a tester. This kl divergence is intractable for each input text since $\Delta$ can not be exactly computed, as you also mentioned. Instead, we use it as a theoretical guide to help us empirically detect text.\
> As we mentioned, the $\Delta$ is relevant to different training mechanisms, data, model size, model architecture, alignment, etc. Thus, inherited from the DetectGPT hypothesis, we assume such a gap is large enough for us to distinguish between human and AI-written text. But in practice, we have to turn to alternative calculations, such as our proposed Bscore and Wscore, showing impressive performance over previous baselines, as reported in Table 1. Based on our experimental observation, like Figure 2, and the outperforming performance in Table 1, we believe our hypothesis is reasonable since it can explain our observed empirical results. For example, the detection effectiveness consistently improves as we increase the number of re-generations $N$ in Figure 4, validating our theoretical result in A.2 PRINCIPLED CHOICE OF K. On the other hand, if we can directly compute the kl divergence, we can perfectly detect any text without needing any detection algorithm. But it is impossible. Thus, we believe our method is solid and effective both empirically and theoretically compared with previous baselines. This is also admitted by reviewer 4FGJ: "The proposed method is simple and effective", reviewer UDX7: "The results are strong" and your kind comment "a simple and neat technique".\
> So, we believe our theoretical and empirical results echo harmoniously, providing a timely solution to the urgent issue of text detection. This is particularly suitable for our ICLR submission track of societal considerations including fairness, safety, privacy.

---

> ### Comment · Reviewer_FS9z · 2023-11-19
> **some followup questions**
>
> I see. I still have some conceptual/intuitive questions and technical questions related to KL and TV discussions.
>
> Let me try to understand the implication of each of your statements:
>
> 1. Bottom of page 3: your log likelihood hypothesis claims that the entropy of machine generated text is smaller than human generated text, which I think is intuitive and captures the observation that machine generated texts are more deterministic.
> 2. Page 4: then you move to prove that both KL and TV should be sufficiently large. So I start to get confused here: (2a)You proved TV and KL of two distributions are large. Here is my confusion: so long as two distributions are sufficiently different, then KL and TV should always be large. Does that mean the assumption you need is not loglikelihood gap hypothesis, but a simpler assumption that distributions of machine generated text and that of human generated text are sufficiently different. (2b) A technical question: your second inequality (moves from difference between two expectations to TV): how is p(\cdot |X) defined? I feel the p(\cdot |X) in two terms are two different functions (representing the conditional pdf for M and H respectively) so you cannot pull out the “shared term” like you did.
>
> Technicality (2b) is always fixable but my central question is what math story you are trying to highlight from these a few lines. I also notice that you pull out KL from TV, which means you really only need two distributions to be different. Llh gap is a sufficient condition but not something you leveraged in your test?
>
> Also, how are BScores and WScores related to KL if you claim KL are intractable? I thought those scores are some best effort to approximate KL but then I feel less sure either now.
>
> My concern is not primarily in the efficacy side but on the explanation side, which I feel is the weaker part of the submission.

---

> > ### Comment · Reviewer_FS9z · 2023-11-19
> > **An additional followup**
> >
> > Or perhaps I can paraphrase the questions:
> >
> > Your logic seems to be following this chain:
> >
> > If this is from human, there is a positive llh gap, then that implies KL divergence is large, then I can design some scoring heuristics that outputs YES when KL divergence is large.
> >
> > (or correct me if my understanding is still inaccurate).
> > My questions can be summarized into two parts:
> > 1. Large KL divergence does not imply a positive llh gap unless you have the other side of inequality, so it does not imply it is from human either (theoretically).
> > 2. Large KL does not need to rely on positive llh gap; it only requires that two distributions are sufficiently different.  So llh gap does not feel like a crucial ingredient to build the theory.

---

> ### Author Response · Authors · 2023-11-20
> **Additional respoonse to "some followup questions and An additional followup"**
>
> **Response**: We would like to thank the reviewer for the additional questions, which are very good suggestions to help improve our work.
>
> First of all, we would like to **clarify a major misunderstanding that we are not designing scoring heuristics to estimate KL divergence that outputs YES when KL divergence is large.** In our previous response, we also clarified that the kl divergence is not a tester.\
> Please see below for more explanations.
>
> 1. Yes, we agree with your valuable observation that a large KL divergence does not imply a positive llh gap unless you have the other side of inequality.\
> Our WScore or BScore is not designed to calculate KL divergence since the human distribution is unknown. Instead, it is used to estimate the $\Delta$. However, since human distribution is intractable in the llh gap, WSocre or BScore is still a rough estimate for $\Delta$. Empirical results demonstrate it is very effective, though theoretically, it is not a 100% exact estimate.
>
> 2. Yes, we agree with your valuable observation that large KL only requires that two distributions are sufficiently different. Our intent is to show that large $\Delta$ helps lead to large KL since KL is not directly calculable.
>
> We re-summarize our intents: First, we assume the human and machine distribution is sufficiently different because otherwise, we can not detect it if they are the same. However, this KL is intractable. Instead, we turn to alternative estimates: large $\Delta$ leads to large KL. Then, our core contribution is the design of the BScore and WScore to estimate $\Delta$ roughly. Empirical results show that those two scores are simple and effective for real-world detection.
>
> We hope our explanation helps explain your concerns.
>
> Thanks,\
> Authors

---

> > ### Comment · Reviewer_FS9z · 2023-11-20
> > **continue discuss**
> >
> > I see. So there are actually two threads:
> >
> > 1. Positive llh gap implies large kl, large tv, so they are statistically distinguishable.
> > 2. Llh gap is q log p - p log p (q human distribution p from machine), where as kl is p log p - p log q, or q log q - q log p. You have blackbox/whitebox access to LLM so you believe either estimating q log p is easier than p log q, or q log q is easier than p log p. Your scores are designed to estimate/approximate/capture llh gap, not kl.

---

> > > ### Author Response · Authors · 2023-11-20
> > > **Thanks!**
> > >
> > > Yes, exactly. Thanks for the suggestion! We will make this more understandable in our revision.

---

### Official Review · Reviewer_UDX7 · 2023-11-01

**Soundness:** 3 good
**Presentation:** 3 good
**Contribution:** 2 fair
**Rating:** 6
**Confidence:** 3

**Summary:**

The paper proposes a training-free detection strategy for identifying machine-generated text by an LLM from human-generated text. The method involves truncating the text and feeding the second half to an LLM and analysing the parts through an N-gram analysis. The paper addresses both black-box (probability distribution of LLMs is not known) as well as white-box models (probability distribution is known). The paper shows that the proposed method surpasses SOA performance in distinguishing between machine and human-generated text by evaluating on a number of proprietary (e.g. GPT3+) and open-source(LLAMA, GPT-NeoX) models, on both English and German datasets. The performance is based on the finding that the distribution of machine-generated text and human-generated text are different when given a preceding text. The proposed method also provides explanation and evidence for the prediction, can handle model sourcing and revised text attacks.

**Strengths:**

- **Significance**: The paper tackles an interesting, relevant problem
- **Originality**: The proposed mathematical framework used is original
- **Results**: The results are strong, showing performance improvement over similar detection methods, like GPTZero and OpenAI classifier
- **Presentation** : The paper has good presentation, and claims follow clearly from experiments

**Weaknesses:**

- The entire framework is based on the likelihood-gap hypothesis. What is the basis for this assumption? Is this formally acknowledged somewhere else in the literature or verified empirically in this paper (e.g. from Fig 2)?
- The paper mentions that is able to provide explanations and evidence in the form of n-grams overlaps. Would be good to provide some examples.

**Questions:**

- Asked above

---

> ### Author Response · Authors · 2023-11-19
> **Author Response**
>
> We would like to thank the reviewer for the positive feedback on our work. Below are our responses.
>
> 1. The entire framework is based on the likelihood-gap hypothesis. What is the basis for this assumption? Is this formally acknowledged somewhere else in the literature or verified empirically in this paper (e.g. from Fig 2)?
>
> **Response**: We have both theoretical and experimental bases for this hypothesis.\
> **Theoretically**: In the baseline work of DetectGPT [1] published on ICML 2023, they also have a likelihood-gap hypothesis (on page 3, section 4): the perturbation discrepancy distance of the machine is positive and large, while it is close to 0 for humans, which means that there is always a positive gap between the human and machine. Therefore, our hypothesis is inherited from theirs.
>
> **Experimentally**: It is also empirically verified by Fig 2, where we can see a clear gap between those two distributions of human and machine. Additionally, our comprehensive experiments in Tab 1 further confirm it by showing that our method achieves SOTA results on 4 datasets and 3 models from OpenAI in both black- and white-box scenarios on 16/16 cases regarding AUROC and 13/16 cases regarding TPR.
>
> [1] Mitchell, Eric, Yoonho Lee, Alexander Khazatsky, Christopher D. Manning, and Chelsea Finn. "Detectgpt: Zero-shot machine-generated text detection using probability curvature." (ICML 2023).
>
> 2. The paper mentions that is able to provide explanations and evidence in the form of n-grams overlaps. Would be good to provide some examples.
>
> **Response**: **We indeed provided examples in our original submission** since this is the unique advantage of our detector over other baselines. On page 8, we had a whole paragraph titled ‘Explainability’ to talk about the explanations and evidence. In Appendix D, we also had the additional tables 14, 15, and 16 for explanations and evidence.\
> Specifically, one example in Figure 1 shows that the typical overlapped n-grams are:
> The original $y_0$ contains the following sentence: “le analyses that focus on specific neighborhoods or regions may reveal disparities that are not apparent in larger-scale analyses. Therefore, it is important to consider the scale of analysis when”. Among 20 re-generations, we found the following two typical examples with significant overlapped n-grams:\
> In $y_1$, the overlapped n-gram is “le analyses that focus on specific neighborhoods or regions may reveal disparities that are not apparent in larger-scale analyses“\
> In $y_5$, the overlapped n-gram is “may reveal disparities that are not apparent in larger-scale analyses. Therefore, it is important to consider the scale of analysis when”\
> We hope this example of overlapped n-grams helps. More examples can be found in Appendix D.
>
> Additionally, we also noticed that on **the OpenAI dev day, the former CEO of OpenAI announced [2] that they would give token logits access on GPT-4 API calls in the future. This will make our white-box detection method also applicable to GPT-4! Although we have not got access to GPT-4 output token logits, we believe this will only make our method more powerful in the future.**
>
> [2] https://openai.com/blog/new-models-and-developer-products-announced-at-devday

---

### Official Review · Reviewer_4FGJ · 2023-11-03

**Soundness:** 4 excellent
**Presentation:** 4 excellent
**Contribution:** 4 excellent
**Rating:** 8
**Confidence:** 4

**Summary:**

This paper studies the problem of detecting whether a piece of text is generated by human or AI models. The authors proposed a simple method to detect the origin of the text, together with two score functions, for both black-box and white-box detection. The authors conducted extensive experiments, and show their approach outperform the state-of-the-art detecting methods.

After rebuttal: I have read the rebuttal and would like to keep my scores.

**Strengths:**

The paper is generally well-written and easy-to-follow. The method proposed by the authors is conducted in a training-free fashion, which remove the need of training detection classifier on millions of texts. The proposed method is simple and effective: based on the authors evaluation, it outperforms many existing detecting methods.

**Weaknesses:**

While the proposed method is effective, it not clear how certain terms in the score function is selected---e.g., the f(n) function in BScore. Can authors comment more on that?

Also, can authors explain a bit more on how P(Y_0 | X) is calculated when Y_0 is human-written text? Does text-davinci-003 allow such calculation?

**Questions:**

Please see above.

---

> ### Author Response · Authors · 2023-11-19
> **Author Response**
>
> We would like to thank the reviewer for the positive feedback on our work. Below are our responses.
>
> 1. While the proposed method is effective, it not clear how certain terms in the score function is selected---e.g., the f(n) function in BScore. Can authors comment more on that?
>
> **Response**: For BScore, we aim to make it simple and effective. The basic idea is to assign higher weights to overlapped N-grams with larger N since it would be more unusual for the machine to generate overlapped N-grams with larger N. As we wrote, “In practice, we set f(n)=n log(n), … More comparisons on parameter sensitivity can be found in Appendix B.” In Appendix B.8, we demonstrate the effects of 6 different weight functions on additional validation sets and found that f(n)=nlog(n) works well across 2 models and 3 datasets. For other terms like the starting or ending N-grams, we follow the same design principle to ensure it is simple and effective.
> Thus, we keep the current choices since they already work well across comprehensive experiments, serving as a strong baseline for future work.
>
> 2. Also, can authors explain a bit more on how P(Y_0 | X) is calculated when Y_0 is human-written text? Does text-davinci-003 allow such calculation?
>
> **Response**: Yes, text-davinci-003 allows such calculations. When Y_0 is human-written text, we could get the logits over Y_0 given X by changing the settings in the OpenAI API: set the max_tokens = 0, n= 1, logprobs=5, echo=True. According to the official documentation [1],
>
> max_tokens = 0: The maximum number of tokens to generate in the completion.
>
> n= 1: How many completions to generate for each prompt.
>
> logprobs=5: Include the log probabilities on the logprobs of the 5 most likely tokens, as well the chosen tokens.
>
> echo=True: Echo back the prompt in addition to the completion
>
> Thus, by the above settings, we were able to get all the token probabilities over Y_0 and X on text-davinci-003.
>
> We also noticed that **on the OpenAI dev day, the former CEO of OpenAI announced [2] that they would give token logits access on GPT-4 API calls in the future. This will make our white-box detection method also applicable to GPT-4! Although we have not got access to GPT-4 output token logits, we believe this will only make our method more powerful in the future**
>
> [1]  https://platform.openai.com/docs/api-reference
>
> [2] https://openai.com/blog/new-models-and-developer-products-announced-at-devday

---

### Meta-Review · Area_Chair_g7Ns · 2023-12-06

**Metareview:**

The paper introduces training-free method for detecting whether a given text is generated by a human or a large language model (LLM). The proposed method, Divergent N-Gram Analysis (DNA-GPT), involves truncating the text and using the preceding portion as input to the LLM to regenerate the remaining parts. The differences between the original and new remaining parts are then analyzed using N-gram analysis or probability divergence. The authors demonstrate that their method outperforms existing detection methods on several datasets and is robust against revised text attacks.

All reviewers acknowledged the relevance and originality of the work. They praised the simplicity and effectiveness of the proposed method, and its ability to outperform existing detection methods. However, some concerns were raised. The major concern expressed about the thoroughness of the study and the limitations of the method when the model used to generate the text is unknown. Although the author claimed they are following the same task-setting as in previous work and they did additional “Model Sourcing” experiments. This still remains an unneglectable issue of this research field. I expect the authors can put more explanations to describe this problem in the future version.

**Justification For Why Not Higher Score:**

1. the reviewers raised valid concerns about the thoroughness of the study and the limitations of the method when the model used to generate the text is unknown. Even though the authors addressed these concerns in their rebuttal, it would be beneficial to see a more comprehensive discussion and potential solutions to these issues in the paper.
2. some aspects of the method, such as the selection of certain terms in the score function and the calculation of certain probabilities, were not fully explained. Providing more detailed explanations and justifications for these choices would strengthen the paper.

**Justification For Why Not Lower Score:**

1. The paper addresses a highly relevant and challenging problem in the field of AI, which is the detection of text generated by large language models. This is a significant contribution to the field.

2. The proposed method, Divergent N-Gram Analysis (DNA-GPT), is original and innovative. Its simplicity also makes it practical.

3. The authors conducted extensive experiments and demonstrated that their method outperforms existing detection methods on several datasets.

---

### Decision · Program_Chairs · 2024-01-16

Accept (poster)